# Supervised Feature Selection with Neuron Evolution in Sparse Neural Networks

**Zahra Atashgahi[1]**      *z.atashgahi@utwente.nl*

**Xuhao Zhang[1]**      *x.zhang-17@student.utwente.nl*

**Neil Kichler[1]**      *neil.kichler@rwth-aachen.de*

**Shiwei Liu[2]**      *s.liu3@tue.nl*

**Lu Yin[2]**      *l.yin@tue.nl*

**Mykola Pechenizkiy[2]**      *m.pechenizkiy@tue.nl*

**Raymond Veldhuis[1]**      *r.n.j.veldhuis@utwente.nl*

**Decebal Constantin Mocanu[1,2]**      *d.c.mocanu@utwente.nl*

[1] *Department of Electrical Engineering, Mathematics and Computer Science, University of Twente, The Netherlands*
[2] *Department of Mathematics and Computer Science, Eindhoven University of Technology, The Netherlands*

**Reviewed on OpenReview:** *https://openreview.net/forum?id=GcO6ugrLKp*

## Abstract

Feature selection that selects an informative subset of variables from data not only enhances the model interpretability and performance but also alleviates the resource demands. Recently, there has been growing attention on feature selection using neural networks. However, existing methods usually suffer from high computational costs when applied to high-dimensional datasets. In this paper, inspired by evolution processes, we propose a novel resource-efficient supervised feature selection method using sparse neural networks, named "NeuroFS". By gradually pruning the uninformative features from the input layer of a sparse neural network trained from scratch, NeuroFS derives an informative subset of features efficiently. By performing several experiments on 11 low and high-dimensional real-world benchmarks of different types, we demonstrate that NeuroFS achieves the highest ranking-based score among the considered state-of-the-art supervised feature selection models. The code is available on GitHub[1].

## 1 Introduction

Feature selection has been gaining increasing importance due to the growing amount of big data. The high dimensionality of data can give rise to issues such as the curse of dimensionality, over-fitting, and high memory and computation demands Li et al. (2018). By removing the irrelevant and redundant attributes in a dataset, feature selection combats these issues while increasing data interpretability and potentially improving the accuracy Chandrashekar & Sahin (2014).

The literature on feature selection can be stratified into three major categories: filter, wrapper, and embedded methods Chandrashekar & Sahin (2014). Unlike filter methods that perform feature selection before the learning task and wrapper methods that use a learning algorithm to evaluate a subset of the features, embedded methods use learning algorithms to determine the informative features Zhang et al. (2019). Since embedded

---

[1] https://github.com/zahraatashgahi/NeuroFS

methods combine feature selection and the learning task into a unified problem, they usually perform more effectively than the other two categories in terms of the quality of the selected features Han et al. (2018); Balın et al. (2019). Therefore, this paper focuses on embedded feature selection due to its superior performance.

In recent years, there has been a growing interest in using artificial neural networks (ANNs) to perform embedded feature selection. This is due to their favorable characteristic of automatically exploring non-linear dependencies among input features, which is often neglected in traditional embedded feature selection methods Tibshirani (1996). In addition, the performance of ANNs scales with the dataset size Hestness et al. (2017), while most feature selection methods do not scale well on large datasets Li et al. (2018). Moreover, many works have demonstrated the efficacy of neural network-based feature selection in both supervised Lu et al. (2018); Lemhadri et al. (2021); Yamada et al. (2020); Singh & Yamada (2020); Wojtas & Chen (2020) and unsupervised Han et al. (2018); Chandra & Sharma (2015); Balın et al. (2019); Doquet & Sebag (2019); Atashgahi et al. (2021); Shaham et al. (2022) settings.

However, while being effective in terms of the quality of the selected features, feature selection with ANNs is still a challenging task. Over-parameterization of neural networks results in high-computational and memory costs, which make their deployment and training on low-resource devices infeasible Hoefler et al. (2021). Only very few works have tried to increase the scalability of feature selection using neural networks on low-resource devices. E.g., Atashgahi et al. (2021) proposes, for the first time, that sparse neural networks Hoefler et al. (2021) can be exploited to perform efficient feature selection. Their proposed method, QuickSelection, which is designed for unsupervised feature selection, trains a sparse neural network from scratch to derive the ranking of the features using the information of the corresponding neurons in the neural network.

In this paper, by introducing dynamic input neurons evolution into the training of a sparse neural network, we propose to use the sparse neural networks to perform supervised feature selection and introduce an efficient feature selection method, named **F**eature **S**election with **Neuro**n Evolution (**NeuroFS**). Our contributions can be summarized as follows:

- We introduce dynamic neuron pruning and regrowing in the input layer of sparse neural networks during training.
- Based on the newly introduced dynamic neuron updating process, we propose a novel efficient supervised feature selection algorithm named "NeuroFS".
- We evaluate NeuroFS on 11 real-world benchmarks for feature selection and demonstrate that NeuroFS achieves the highest average ranking among the considered feature selection methods on low and high-dimensional datasets.

## 2 Background

In this section, we provide background information on feature selection and sparse neural networks.

### 2.1 Feature Selection

#### 2.1.1 Problem Formulation

In this section, we first describe the general supervised feature selection problem. Consider a dataset $\mathbb{X}$ containing $m$ samples $(\boldsymbol{x}^{(i)}, y^{(i)})$, where $\boldsymbol{x}^{(i)} \in \mathbb{R}^d$ is the $i$-th sample in data matrix $\boldsymbol{X} \in \mathbb{R}^{m \times d}$, $d$ is the dimensionality of the dataset or the number of the features, and $y^{(i)}$ is the corresponding label for supervised learning. Feature selection aims to select a subset of the most discriminative and informative features of $\boldsymbol{X}$ as $\mathbb{F}_s \subset \mathbb{F}$ such that $|\mathbb{F}_s| = K$, where $\mathbb{F}$ is the original feature set, and $K$ is a hyperparameter of the algorithm which indicates the number of features to be selected.

**Objective function**: In supervised feature selection, we seek to optimize:

$$\mathbb{F}_s^* = \operatorname*{arg\,min}_{\mathbb{F}_s \subset \mathbb{F}, |\mathbb{F}_s| = K} \sum_{i=0}^{m} J(f(\boldsymbol{x}_{\mathbb{F}_s}^{(i)}; \boldsymbol{\theta}), y^{(i)}), \tag{1}$$

where $\mathbb{F}_s^*$ is the final selected feature set, $J$ is a desired loss function, and $f(\boldsymbol{x}_{\mathbb{F}_s}^{(i)}; \boldsymbol{\theta})$ is a classification function parameterized by $\boldsymbol{\theta}$ aiming at estimating the target for the $i$-th sample using a subset of features $\boldsymbol{x}_{\mathbb{F}_s}^{(i)}$.

Solving this optimization problem can be a challenging task. As the choice of feature subset $\mathbb{F}_s$ grows exponentially with increasing number of features $d$, solving Equation 1 is a NP-hard problem. Additionally, it is important that function $f$ that can learn a fruitful representation and complex data dependencies Lemhadri et al. (2021). We choose artificial neural networks due to their high expressive power; a simple one-hidden layer feed-forward neural network is known to be a universal approximator Goodfellow et al. (2016). Finally, as we aim to select features in a computationally efficient manner, in this paper, we choose sparse neural networks to represent the data and perform feature selection.

### 2.1.2 Related Work

Feature selection methods are classified into three main categories: filter, wrapper, and embedded methods. **Filter methods** use criteria such as correlation Guyon & Elisseeff (2003), mutual information Chandrashekar & Sahin (2014), Laplacian score He et al. (2006), to rank the features independently from the learning task, which makes them fast and efficient. However, they are prone to selecting redundant features Chandrashekar & Sahin (2014). **Wrapper methods** find a subset of features that maximize an objective function Zhang et al. (2019) using various search strategies such as tree structures Kohavi & John (1997) and evolutionary algorithms Liu et al. (1996). However, these methods are costly in terms of computation. **Embedded methods** aim to address the drawbacks of the filter and wrapper approaches by integrating feature selection and training tasks to optimize the subset of features. Various approaches have been used to perform embedded feature selection including, mutual information Battiti (1994); Peng et al. (2005), the SVM classifier Guyon et al. (2002), and neural networks Setiono & Liu (1997).

Recently, neural network-based feature selection in both supervised Lu et al. (2018); Lemhadri et al. (2021); Yamada et al. (2020); Singh & Yamada (2020); Wojtas & Chen (2020) and unsupervised Atashgahi et al. (2021); Balın et al. (2019); Han et al. (2018); Chandra & Sharma (2015); Doquet & Sebag (2019) settings have gained increasing attention due to their favorable advantages of capturing non-linear dependencies and showing good performance on large datasets. However, most of these methods suffer from over-parameterization, which leads to high computational costs, particularly on high-dimensional datasets. QuickSelection Atashgahi et al. (2021) addresses this issue by exploiting sparse neural networks; however, due to the random growth of connections in its topology update stage, it might not be able to detect fastly enough the informative features on high-dimensional datasets due to the large search space. As we show in the following sections, we address this issue by gradually pruning uninformative input neurons and exploiting gradients to speed up the learning process.

## 2.2 Sparse Neural Networks

Sparse neural networks have been proposed to address the high computational costs of dense neural networks Hoefler et al. (2021). They aim to reduce the parameters of a dense neural network while preserving a decent level of performance on the task of interest.

There are two main approaches to obtain a sparse neural network: dense-to-sparse and sparse-to-sparse methods Mocanu et al. (2021).
**Dense-to-sparse** algorithms start with a dense network and prune the unimportant connections to obtain a sparse network LeCun et al. (1990); Hassibi & Stork (1993); Han et al. (2015); Lee et al. (2019); Frankle & Carbin (2018); Molchanov et al. (2017; 2019); Gale et al. (2019). As they start with a dense network, they need the memory and computational resources to fit and train the dense network for at least a couple of iterations. Therefore, they are mostly efficient during the inference phase.
**Sparse-to-sparse** algorithms aim to bring computational efficiency both during the training and inference. These methods use a static Mocanu et al. (2016) or dynamic Mocanu et al. (2018); Bellec et al. (2018) sparsity pattern during training. In the following, we will elaborate on sparse training with dynamic sparsity (or started to be known in the literature as dynamic sparse training (DST)), which usually outperforms the static approach.

### 2.2.1 Dynamic Sparse Training (DST)

DST is a class of methods to train sparse neural networks sparsely from scratch. DST methods aim at optimizing the sparse connectivity of a sparse neural network during training, such that they never use dense network matrices during training Mocanu et al. (2021). Formally, DST methods start with a sparse neural network $f(\boldsymbol{x}, \boldsymbol{\theta}_s)$ with a sparsity level of $S$. We have $S = 1 - \frac{\|\boldsymbol{\theta}_s\|_0}{\|\boldsymbol{\theta}\|_0}$, where $\boldsymbol{\theta}_s$ is a subset of parameters of the equivalent dense network parameterized by $\boldsymbol{\theta}$, $\|\boldsymbol{\theta}_s\|_0$ and $\|\boldsymbol{\theta}\|_0$ are the number of parameters of the sparse and dense network, respectively. They aim to optimize the following problem:

$$\boldsymbol{\theta}_s^* = \underset{\boldsymbol{\theta}_s \in \mathbb{R}^{\|\boldsymbol{\theta}\|_0}, \, \|\boldsymbol{\theta}_s\|_0 = D\|\boldsymbol{\theta}\|_0}{\arg\min} \frac{1}{m} \sum_{i=1}^{m} J(f(\boldsymbol{x}^{(i)}; \boldsymbol{\theta}_s), \boldsymbol{y}^{(i)}), \tag{2}$$

where $D = 1 - S$ is called density level. During training, DST methods periodically update the sparse connectivity of the network; e.g., in Mocanu et al. (2018); Evci et al. (2020) authors remove a fraction $\zeta$ of the parameters $\boldsymbol{\theta}_s$ and add the same number of parameters to the network to keep the sparsity level fixed. In the literature, usually, weight magnitude has been used as a criterion for dropping the connections. However, there exists various approaches for weight regrowth including, random Mocanu et al. (2018); Mostafa & Wang (2019), gradient-based Evci et al. (2020); Dai et al. (2019); Dettmers & Zettlemoyer (2019); Jayakumar et al. (2020), locality-based Hoefler et al. (2021), and similarity-based Atashgahi et al. (2019). It has been shown that in many cases, they can match or even outperform their dense counterparts Frankle & Carbin (2018); Mocanu et al. (2018); Liu et al. (2021a;b). Evci et al. (2022) have discussed in-depth that DST methods improve the gradient flow in the network by updating the sparse connectivity that eventually leads to a good performance. In this paper, we exploit sparse neural network training from scratch to design an efficient supervised feature selection method.

## 3 Proposed Method

In this section, we present our proposed methodology for feature selection using sparse neural networks, named **F**eature **S**election with **Neuro**n evolution (**NeuroFS**). We start by describing our proposed sparse training algorithm. Then, we explain how the introduced sparse training algorithm can be used to perform feature selection.

### 3.1 Dynamic Neuron Evolution

Inspired by the weights update policy in DST, we introduce dynamic neuron evolution in the framework of DST to perform efficient feature selection. While existing DST methods update only the connections or the hidden neurons Dai et al. (2019) to evolve the topology of sparse neural networks, we propose to update also the input neurons of the network to dynamically derive a set of relevant features of the given input data.

Our proposed neuron evolution process has two steps. Consider a network in which only a fraction of input neurons have non-zero connections. We periodically update the input layer connectivity by first dropping a fraction of unimportant neurons (*neuron removal*) and then adding a number of unconnected neurons back to the network (*neuron regrowth*):

**Neuron Removal.** The criterion used for dropping the neurons is strength, which is introduced in Atashgahi et al. (2021). Strength is the summation of the absolute weights of existing connections for an input neuron. A higher strength of a neuron indicates that the corresponding input feature has higher importance in the data. Therefore, we drop a fraction of low-strength neurons at each epoch. We call the neurons with at least one non-zero weight connection, *active*, and the neurons without any non-zero connections, *inactive*.

**Neuron Regrowth.** After removing unimportant neurons, we explore the inactive neurons. We activate a number of neurons with the highest potential to enhance the learned data representation. We exploit the gradient magnitude of the non-existing connections for each neuron as a criterion to choose the most important inactive neurons. It has been shown in Evci et al. (2020) that adding the zero-connections with the largest gradients magnitude in the DST process accelerates the learning and improves the accuracy. Evci

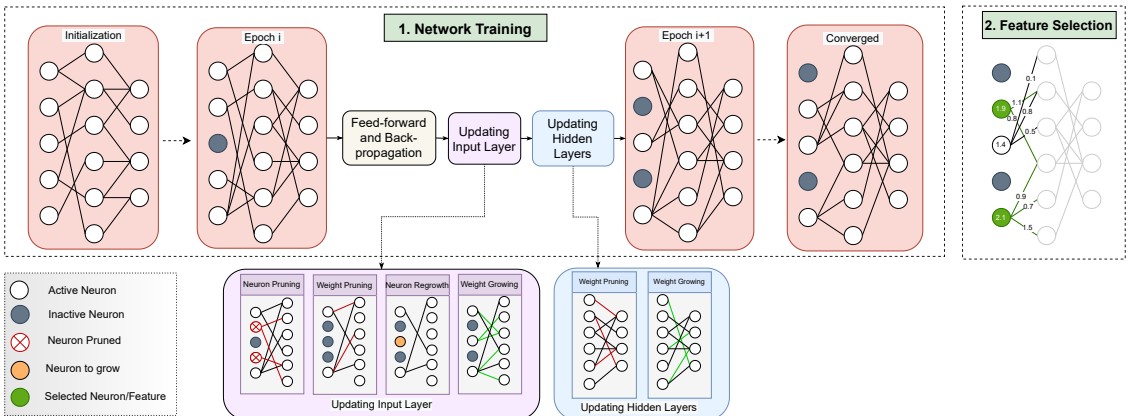

Figure 1: Overview of "NeuroFS". At initialization, a sparse MLP is initialized (Section 3.2.1). During training, at each training epoch, after standard feed-forward and back-propagation, input and hidden layers are updated such that a large fraction of unimportant input neurons are gradually dropped while giving a chance to the removed input neurons for regrowth (Section 3.2.2). After convergence, NeuroFS selects the corresponding features to $K$ active neurons with the highest strength (Section 3.2.3).

et al. (2022) also have shown that picking the connections with the highest gradient magnitude increases the gradient flow, which eventually leads to a decent performance. We hypothesize that adding inactive neurons connected to the zero-connections with the highest gradient magnitude to the network would improve the data representation and increase the likelihood of finding an informative set of features.

Dynamic neuron evolution is loosely inspired by evolutionary algorithms Stanley & Miikkulainen (2002). Still, due to the large search space, the latter cannot be directly applied to our problem without significantly increased computational time. To alleviate this, we seek inspiration in the dynamics of the evolution process from the biological brain at the epigenetic level, which performs cellular changes (seconds to days time scale) Kowaliw et al. (2014), and not at the phylogenic level (generations time scale) as it is usually performed in evolutionary computing. Accordingly, NeuroFS removes and regrows neurons in the input layer of a sparsely trained neural network based on chosen criteria at each epoch until a reduced optimal set of input neurons remains active in the network. In the next section, we will explain how NeuroFS uses dynamic neuron evolution to perform feature selection.

## 3.2 NeuroFS

Our proposed algorithm is briefly sketched in Figure 1. In short, NeuroFS aims at efficiently selecting a subset of features that can learn an effective representation of the input data in a sparse neural network. In the following, we describe the algorithm in more details.

### 3.2.1 Problem Setup

We first start by describing the network structure and problem setup.

**Network Architecture.** We exploit a supervised deep neural network, Multi-Layer Perceptron (MLP). We initialize a sparse MLP $f(\boldsymbol{x}, \boldsymbol{\theta}_s)$, with $L$ layers and sparsity level of $S$.

**Initialization.** The sparse connectivity is initialized randomly as an Erdos-Renyi random graph Mocanu et al. (2018). Sparsity level $S$ is determined by a hyperparameter of the model, named $\varepsilon$, such that the density of layer $l$ is $\varepsilon(n^{(l-1)}+n^{(l)})/(n^{(l-1)} \times n^{(l)})$, and the total number of parameters is equal to $\|\boldsymbol{\theta}_s\|_0 = \sum_{l=1}^{L} \|\boldsymbol{\theta}_s^{(l)}\|_0$, where $l \in \{1, 2, ..., L\}$ is the layer index and $n^{(l)}$ is number of neurons at layer $l$. The number of connections at each layer is computed as $\|\boldsymbol{\theta}_s^{(l)}\|_0 = \varepsilon(n^{(l-1)}+n^{(l)})$.

### 3.2.2 Training

After initializing the network, we start the training process. In summary, we start with a sparse neural network and aim to optimize the topology of the network and the selected subset of features simultaneously. During training, we gradually remove the input neurons while giving a chance for the inactive neurons to

be re-added to the network. Finally, when the training is finished, we select the important features from a limited number of active neurons. In the following, we describe the training algorithm in more detail.

At each training epoch, NeuroFS performs the following three steps:

*1. Feed-forward and Back-propagation.* At each epoch, first, standard feed-forward and back-propagation are performed to train the weights of the sparse neural network.

*2. Updating Input Layer.* After each training epoch, we update the input layer. The novelty of our proposed algorithm lies mainly in updating the input layer. During training, NeuroFS gradually decreases the number of active input features. In short, at epoch $t$, it gradually prunes a number of input neurons ($c_{prune}^{(t)}$) and regrows a number of unconnected neurons ($c_{grow}^{(t)}$) back to the network. Updating the input layer in NeuroFS consists of two phases:

- **Removal Phase.** From the beginning of the training until $t_{removal}$, updating the input layer is at the removal phase. In this phase, the total number of active neurons decreases at each epoch such that $c_{prune}^{(t)} > c_{grow}^{(t)}$, if $t \leqslant t_{removal}$. We have $t_{removal} = \lceil \alpha t_{max} \rceil$, where $0 < \alpha < 1$ is a hyperparameter of NeuroFS determining the neuron removal phase duration, $\lceil \rceil$ is the ceiling function, and $t_{max}$ is the total number of epochs.
- **Update Phase.** From $t_{removal}$ until the end of training, the number of connected neurons remains fixed in the network and only a fraction of neurons are updated. In other words, $c_{prune}^{(t)} = c_{grow}^{(t)}$, if $t > t_{removal}$.

Formally, we compute $c_{prune}^{(t)}$ at epoch ($t$) as follows:

$$
c_{prune}^{(t)} = \begin{cases} c_{remove}^{(t)} + c_{grow}^{(t)}, & t \leqslant t_{removal} \\ c_{grow}^{(t)}, & \text{otherwise} \end{cases}. \tag{3}
$$

$c_{prune}^{(t)}$ in the removal phase consists of two parts: $c_{remove}^{(t)}$ and $c_{grow}^{(t)}$. As the overall number of active neurons is decreasing in this phase, $c_{remove}^{(t)}$ extra neurons to the updated ones will be removed at each epoch. $c_{remove}^{(t)}$ is computed as:

---

**Algorithm 1** NeuroFS

**Input**: Dataset $\mathbb{X}$, sparsity hyperparameter $\varepsilon$, drop fractions $\zeta_{in}$ and $\zeta_h$, neuron removal phase duration hyperparameter $\alpha$, number of training epochs $t_{max}$, number of features to select $K$.
**Initialization**: Initialize the network with sparsity level $S$ determined by $\varepsilon$ (Section 3.2.1)
**for** $t \in \{1, \ldots, \#t_{max}\}$ **do**
  **I. Standard feed-forward and back-propagation**
  **II. Update Input Layer:**
    0. Compute $c_{prune}^{(t)}$ (Equation 3) and $c_{grow}^{(t)}$ (Equation 7).
    1. Drop $c_{prune}^{(t)}$ neurons with the lowest strength.
    2. Drop a fraction $\zeta_{in}$ of connections with the lowest magnitude.
    3. Select $c_{grow}^{(t)}$ inactive neurons (that have connections with the highest gradient magnitude), to be activated.
    4. Regrow as many connections as have been removed to the active neurons.
  **III. Update Hidden Layers:**
    **for** $l \in \{1, \ldots, L\}$ **do**
      1. Drop a fraction $\zeta_h$ of connections with the lowest magnitude from layer $h^l$.
      2. Regrow as many connections as have been removed in layer $h^l$.
    **end for**
**end for**
**Feature Selection:**
  Select $K$ features corresponding to the active neurons with the highest strength in the input layer.

---

$$c_{remove}^{(t)} = \lceil \frac{R - R^{(t)}}{t_{removal} - t} \rceil, \tag{4}$$

$$R^{(t)} = \sum_{i=1}^{t-1} c_{remove}^{(i)}, \tag{5}$$

$$R = \lceil (1 - \zeta_{in})d - K \rceil, \tag{6}$$

where $R^{(t)}$ is the total number of inactive neurons at epoch $t$, $R$ is the total number of neurons to be removed in the removal phase, and $\zeta_{in} \in \mathbb{R}, 0 < \zeta_{in} < 1$ is the update fraction of the input layer. In other words, the total number of active neurons after the removal phase is $\zeta_{in}d + K$. We keep $\zeta_{in}d$ neurons extra to the number of selected features $K$, so that the update phase does not disturb the already found important features.

Finally, the number of neurons to grow at epoch $t$ is computed as:

$$c_{grow}^{(t)} = \lceil \zeta_{in}(1 - \frac{t}{t_{max}})R^{(t)} \rceil. \tag{7}$$

In other words, at each epoch, we add a fraction $\zeta_{in}$ of the inactive neurons back to the network. However, as the number of inactive neurons increases during training, the number of updated neurons will increase consequently. A large number of updated neurons might diverge the network training. Therefore, we decrease the update fraction linearly during training. At epoch $t$, we update $\zeta_{in}(1 - \frac{t}{t_{max}})$ proportion of the total inactive neurons.

After computing $c_{prune}^{(t)}$ and $c_{grow}^{(t)}$, the input layer is updated as follows:

1. **Neuron pruning:** $c_{prune}^{(t)}$ neurons with lowest strength are dropped from the input layer. The strength of input neuron $i$ is computed as $s_i = \left\| \boldsymbol{w}^{(i)} \right\|_1$, where $\boldsymbol{w}^{(i)}$ is the weights vector of neuron $i$.
2. **Weight pruning:** a fraction $\zeta_{in}$ of connections with the lowest magnitudes are dropped from the active input features.
3. **Neuron regrowth:** $c_{grow}^{(t)}$ neurons are selected for being activated and added to the network. As discussed in Section 3.1, these neurons are the ones connected to the connections with the largest absolute gradient among all non-existing connections of inactive neurons.
4. **Weight growing:** the same number as the number of removed connections will be added to the network so that the sparsity remains fixed during training. These connections are the ones with the largest absolute gradient among all non-existing connections of the active neurons at the current epoch.

*3. Updating Hidden Layers.* Hidden layers will be updated by updating the sparse connectivity, which is the standard approach in the DST process. We use gradients for weight regrowth Evci et al. (2020). For each hidden layer $h^{(l)}$, NeuroFS performs the following two steps:

1. **Weight pruning:** a fraction $\zeta_h$ of connections with the lowest magnitude are dropped from layer $h^{(l)}$.
2. **Weight growing:** the same number as the number of removed connections will be added to layer $h^{(l)}$. These connections are the ones with the largest absolute gradient among all non-existing connections.

### 3.2.3 Feature Selection

After the training process is finished, we perform feature selection. We select $K$ neurons with the highest strength out of the $\zeta_{in}d + K$ remained active neurons. The corresponding feature to these $K$ neurons are the most informative and relevant features in our dataset. NeuroFS is schematically described in Figure 1 and the corresponding pseudocode is available at Algorithm 1.

## 4 Experiments and Results

In this section, we first describe the experimental settings and then analyze the performance of NeuroFS and compare it with several state-of-the-art feature selection methods.

Table 1: Datasets characteristics.

| Dataset | Type | # Features | # Samples | # Train | # Test | # Classes |
|---|---|---|---|---|---|---|
| COIL-20 | | 1024 | 1440 | 1152 | 288 | 20 |
| USPS | Image | 256 | 9298 | 7438 | 1860 | 10 |
| MNIST | | 784 | 70000 | 60000 | 10000 | 10 |
| Fashion-MNIST | | 784 | 70000 | 60000 | 10000 | 10 |
| Isolet | Speech | 617 | 7737 | 6237 | 1560 | 26 |
| HAR | Time Series | 561 | 10299 | 7352 | 2947 | 6 |
| BASEHOCK | Text | 4862 | 1993 | 1594 | 399 | 2 |
| Arcene | Mass Spectrometry | 10000 | 200 | 160 | 40 | 2 |
| Prostate_GE | | 5966 | 102 | 81 | 21 | 2 |
| SMK-CAN-187 | Biological | 19993 | 187 | 149 | 38 | 2 |
| GLA-BRA-180 | | 49151 | 180 | 144 | 36 | 4 |

## 4.1 Settings

This section describes the experimental settings, including, datasets, compared methods, hyperparameters, implementation, and the evaluation metric.

**Datasets.** We evaluate the effectiveness of NeuroFS on eleven datasets[2] described in Table 1.

**Comparison.** We have selected seven state-of-the-art feature selection methods for comparison as follows:

*Embedded methods:* LassoNet Lemhadri et al. (2021) exploits a neural network with residual connections to the input layer and solves a two-component (linear and non-linear) optimization problem to find the feature importance. STG Yamada et al. (2020) exploits a continuous relaxation of Bernoulli distribution in a neural network to perform feature selection. QuickSelection Atashgahi et al. (2021) (denoted as QS in the Figures) selects features using the strength of input neurons of a sparse neural network. RFS Nie et al. (2010) employs a joint $\ell_{2,1}$-norm minimization on the loss function and regularization to select features.

*Filter methods:* Fisher_score Gu et al. (2011) selects features that maximizes similarity of feature values among the same class. CIFE Lin & Tang (2006) maximizes the conditional redundancy between unselected and selected features given the class labels. Finally, ICAP Jakulin (2005) iteratively selects features maximizing the mutual information with the class labels given the selected features.

**Hyperparameters.** The architecture of the network used in the experiments is a 3-layer sparse MLP with 1000 neurons in each hidden layer. The activation function used for the hidden layers is *Tanh* (except for Isolet dataset where *Relu* is used), and the output layer activation function is *Softmax*. The values for the hyperparameters were found through a grid search among a small set of values. We have used stochastic gradient descent (SGD) with a momentum of 0.9 as the optimizer. The parameters for training neural network-based methods, including batch size, learning rate, and the number of epochs ($t_{max}$), have been set to 100, 0.01, and 100, respectively. However, the batch size for datasets with few samples ($m \leq 200$) was set to 20. The hyperparameter determining the sparsity level $\varepsilon$ is set to 30. Update fraction for the input layer $\zeta_{in}$ and hidden layer $\zeta_h$ have been set to 0.2 and 0.3 respectively. Neuron removal duration hyperparameter $\alpha$ is set to 0.65. $\zeta_{in}$ and $\alpha$ are the only hyperparameters particular to NeuroFS. We use min-max scaling for data preprocessing for all methods except for the BASEHOCK dataset, where we perform standard scaling with zero mean and unit variance.

**Implementation.** We implemented our proposed method using Keras Chollet et al. (2015). The starting point of our implementation is based on the sparse evolutionary training introduced as SET in Mocanu et al. (2018)[3] to which we added the gradient-based connections growth proposed in RigL Evci et al. (2020). For Fisher_score, CIFE, ICAP, and RFS, we have used the implementations provided by the *Scikit-Feature* library Li et al. (2018)[4]. The hyperparameter of RFS ($\gamma$) has been set to 10 (searched among $[0.01, 0.1, 0.5, 1, 10]$). We implemented QuickSelection Atashgahi et al. (2021) in our code; we adapted it to supervised feature selection, as this was not done in the paper proposing QuickSelection. We have used a similar structure and sparsity level ($\epsilon = 30$) to our method for a fair comparison. For QuickSelection, we set $\zeta = 0.3$. For STG and LassoNet, we used the implementation provided by the authors[5,6]. For STG, we used a 3-layer MLP with 1000 hidden neurons in each layer and set the hyperparameter $\lambda = 0.5$ (searched among $[0.001, 0.01, 0.5, 1, 10]$). For LassoNet, we used a 1-layer MLP with 1000 hidden neurons and set $M = 10$, as suggested by the authors Lemhadri et al. (2021). Please note that we have also tried using a 3-layer MLP for LassoNet. However, it significantly

---

[2]Available at `https://jundongl.github.io/scikit-feature/datasets.html`

[3]`https://github.com/dcmocanu/sparse-evolutionary-artificial-neural-networks`

[4]`https://jundongl.github.io/scikit-feature/`

[5]`https://github.com/lasso-net/lassonet`

[6]`https://github.com/runopti/stg`

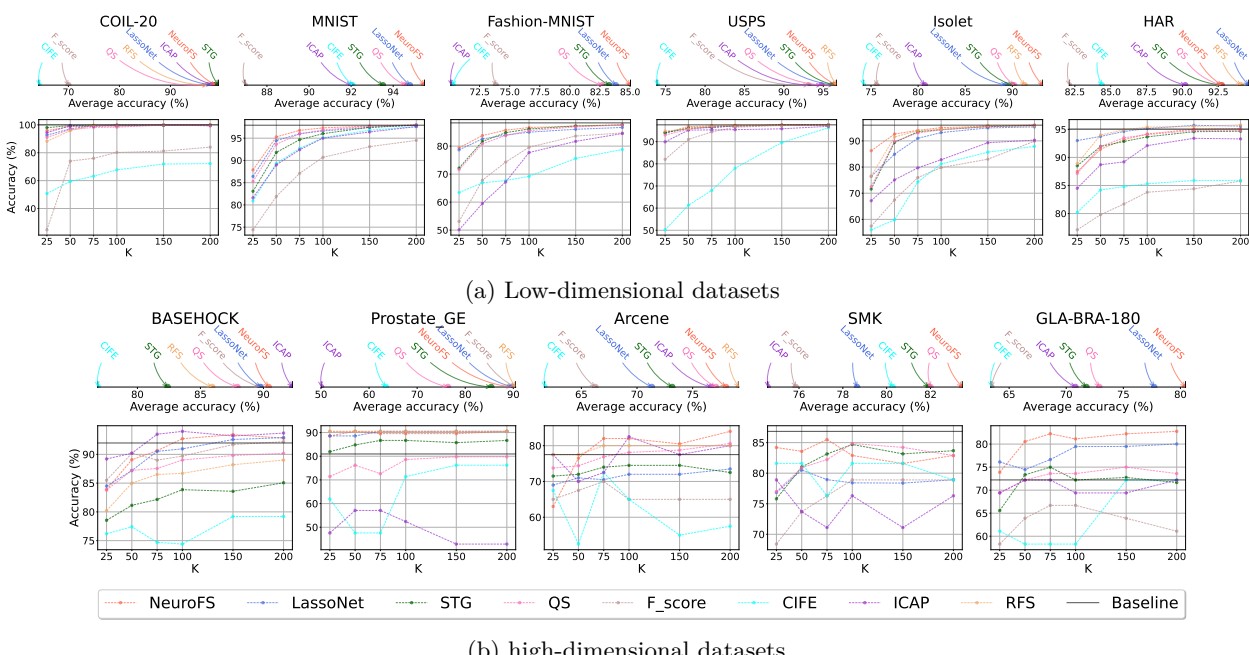

(a) Low-dimensional datasets

(b) high-dimensional datasets

Figure 2: Supervised feature selection comparison for low (a) and high-dimensional (b) datasets, including accuracy for various values of $K$ (below) and average accuracy over $K$ (above).

increased the running time, and particularly on large datasets, it exceeded the 12 hours running time. In addition, in the other cases, it did not lead to significantly different results than LassoNet with 1-layer MLP. To have a fair comparison, for NN-based methods (NeuroFS, LassoNet, STG, and QS), we used similar training hyperparameters, including learning rate (0.01), optimizer (SGD), batch size (100, except 20 for datasets with few samples (m<=200)), and training epoch (100). We consider a 12 hours limit on the running time of each experiment. The results of the experiments that exceed this limit are discarded. We used a *Dell R730* processor to run the experiments. We run neural network-based methods using *Tesla-P100* GPU with 16G memory.

**Evaluation Metrics.** For evaluating the methods, we use classification accuracy of a *SVM* classifier Keerthi et al. (2001) with RBF kernel implemented by Scikit-Learn library[7] and used the default hyperparameters of this library. As some of the compared methods do not exploit neural networks to perform feature selection, we intentionally use a non-neural network-based classifier to ensure that the evaluation process is objective and does not take advantage of the same underlying mechanisms as our method. We first find the $K$ important features using each method. Then, we train a SVM classifier on the selected features subset of the training set. We report the classification accuracy on the test set as a measure of performance. We have also evaluated the methods using two other classifiers including KNN and ExtraTrees in Appendix C. We have considered classification accuracy using all features as the *baseline* method.

---

[7] https://scikit-learn.org/stable/modules/generated/sklearn.svm.SVC.html

Table 2: Supervised feature selection comparison (average classification accuracy over various $K$ values (%)). Empty entries show that the corresponding experiments exceeded the time limit (12 hours). Bold and italic fonts indicate the best and second-best performer, respectively.

| Method | Low-dimensional Datasets | | | | | | High-dimensional Datasets | | | | |
| | COIL-20 | MNIST | Fashion-MNIST | USPS | Isolet | HAR | BASEHOCK | Prostate_GE | Arcene | SMK | GLA-BRA-180 |
|---|---|---|---|---|---|---|---|---|---|---|---|
| Baseline | 100.0 | 97.92 | 88.3 | 97.58 | 96.03 | 95.05 | 91.98 | 80.95 | 77.5 | 86.84 | 72.22 |
| NeuroFS | *98.79±0.22* | **95.48 ± 0.34** | **85.03 ± 0.15** | **96.68 ± 0.16** | **93.22 ± 0.11** | 92.74 ± 0.23 | *90.42±0.80* | 89.70 ± 0.72 | *78.00±1.78* | **82.36 ± 0.98** | **80.46 ± 0.99** |
| LassoNet | 98.03 ± 0.31 | *94.80±0.23* | 83.81±0.12 | 96.41±0.05 | 89.31 ± 0.13 | **94.63 ± 0.10** | 89.77 ± 0.56 | *89.86±0.78* | 71.33 ± 1.94 | 78.67 ± 2.09 | *77.70±1.84* |
| STG | **99.30 ± 0.31** | 94.16 ± 0.47 | 83.74 ± 0.41 | *96.55±0.17* | 89.13 ± 1.43 | 91.87 ± 0.63 | 85.48 ± 0.78 | 85.41 ± 2.72 | 73.75 ± 2.78 | 81.34 ± 2.68 | 71.19 ± 2.33 |
| QS | 97.23 ± 1.34 | 94.57 ± 0.35 | 82.69 ± 0.24 | 96.22 ± 0.20 | 90.20 ± 1.23 | 92.70 ± 0.57 | 87.93 ± 0.40 | 76.39 ± 7.44 | 77.08 ± 1.56 | *82.01±2.69* | 72.91 ± 0.69 |
| Fisher_score | 70.02 ± 0.00 | 86.95 ± 0.00 | 73.85 ± 0.00 | 93.12 ± 0.00 | 75.58 ± 0.00 | 82.10 ± 0.00 | 89.72 ± 0.00 | **90.50 ± 0.00** | 66.25 ± 0.00 | 75.85 ± 0.00 | 63.43 ± 0.00 |
| CIFE | 64.18 ± 0.00 | 92.07 ± 0.00 | 70.27 ± 0.00 | 73.90 ± 0.00 | 74.15 ± 0.00 | 84.38 ± 0.00 | 76.85 ± 0.00 | 63.48 ± 0.00 | 61.67 ± 0.00 | 80.27 ± 0.00 | 63.40 ± 0.00 |
| ICAP | 98.67 ± 0.00 | 92.00 ± 0.00 | 70.12 ± 0.00 | 94.75 ± 0.00 | 80.72 ± 0.00 | 90.20 ± 0.00 | **92.30 ± 0.00** | 50.00 ± 0.00 | 76.67 ± 0.00 | 74.57 ± 0.00 | 70.80 ± 0.00 |
| RFS | 97.28 ± 0.00 | - | - | **96.68 ± 0.00** | *91.32±0.00* | *94.08±0.00* | 85.93 ± 0.00 | **90.50 ± 0.00** | **79.17 ± 0.00** | - | - |

## 4.2 Feature Selection Evaluation

In this section, we evaluate the performance of NeuroFS and compare it with several feature selection algorithms. We run all the methods on the datasets described in Section 4.1 and for several values of $K \in \{25, 50, 75, 100, 150, 200\}$. Then, we evaluate the quality of the selected set of features by measuring the classification accuracy on an unseen test set as described in Section 4.1. The results are an average of five different seeds. The detailed results for low and high-dimensional dataset, including accuracy for various values of $K$ (below) and average accuracy over $K$ (above), are demonstrated in Figure 2. We have also presented the detailed results for each value of $K$ in Table 8 in Appendix E. To summarize the results and have a general overview of the performance of each method independent of a particular $K$ value, we have shown the average accuracy over the different values of $K$ in Table 2.

As presented in Figure 2 and Table 2, NeuroFS is the best performer in 6 datasets out of 11 considered datasets in terms of average accuracy, while performing very closely to the best performer in the remaining cases. Filter methods, such as ICAP, CIFE, and F-score, have been outperformed by embedded methods on most datasets considered, as they select features independently from the learning task. Among these methods, ICAP performs well on the text dataset (BASEHOCK); this can show that mutual information is informative in feature selection from the text datasets. Among the considered embedded methods, RFS fails to find the informative features on datasets with a high number of samples (e.g., MNIST, Fashion-MNIST) or dimensions (e.g., SMK, GLA-BRA-180) within the considered time limit.

By looking into the results of all considered methods, it can be observed that neural network-based feature selection methods outperform classical feature selection methods in most cases. Therefore, it can be concluded that the complex non-linear dependencies extracted by the neural network are beneficial for the feature selection task. However, as will be discussed in Section 5.2, the over-parameterization in dense neural networks, as used for STG and LassoNet, leads to high computational costs and memory requirements, particularly on high-dimensional datasets. NeuroFS and QuickSelection address this issue by exploiting sparse layers instead of dense ones.

NeuroFS outperforms QuickSelection, which is the sparse competitor of NeuroFS, in terms of average accuracy, particularly on the high-dimensional datasets. This is because, for high-dimensional datasets, QuickSelection needs more training time to find the optimal topology in the large connections search space due to the random search. NeuroFS alleviate this problem by exploiting the gradient of the connections to find the informative paths in the network while removing the uninformative neurons gradually to reduce the search space.

To summarize the results and have a general overview of the methods' performance, we use a ranking-based score. For each dataset and value of $K$, we rank the methods based on their classification accuracy and give a score of 0 to the worst performer, and the highest score ($\#methods-1$) to the best performer. For each method, we compute the average score for different values of $K$ and different datasets. The results are summarized in Figure 3. NeuroFS achieves the highest average ranking on both low and high-dimensional datasets.

Overall, it can be concluded that inspired by the evolutionary process, NeuroFS can find an effective subset of features by dynamically changing the sparsity pattern in both input neurons and connections. By dropping the unimportant input neurons (based on magnitude) and adding new neurons based on the incoming gradient, it can mostly outperform its direct competitors, LassoNet, STG, and QuickSelection, in terms of accuracy while being efficient by using sparse layers instead of dense over-parameterized layers.

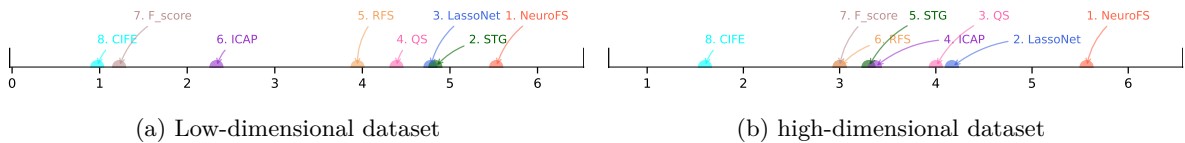

(a) Low-dimensional dataset        (b) high-dimensional dataset

Figure 3: Average ranking score over all datasets and $K$ values.

Figure 4: Feature importance visualization on the MNIST dataset (number of selected features K=50).

### 4.3 Feature Importance Visualization

In order to gain a better understanding of the NeuroFS algorithms, in this section, we analyze the feature importance during the training of the network. We run NeuroFS on the MNIST dataset and for $K = 50$ and visualize the strength of input neurons as a heat-map at several epochs in Figure 4.

As shown in Figure 4, at the initialization, all the neurons have very close strength/importance. This stems from the random initialization of the weights to a small random value. During training, the number of active neurons gradually decreases. The removed neurons are mostly located towards the edges of this picture. This pattern is similar to the MNIST digits dataset, where most digits appear in the middle of the image. Finally, at the last epoch, a limited number of neurons have remained active. We select the most important features out of the active features. In conclusion, this experiment shows that NeuroFS can determine the most important region in the features accurately.

## 5 Discussion

In this section, we present the results of several analyses on the performance of NeuroFS, including robustness evaluation and hyperparameter's effect. We have additionally analyzed weight/neuron growth policy in Appendix A, and compared NeuroFS with two HSICLasso-based feature selection methods and RigL in Appendix B and D, respectively.

### 5.1 Robustness Evaluation: Topology Variation

In this section, we analyze the robustness of NeuroFS to variation in the topology. We aim to explore if different runs of NeuroFS converge to similar or distant topologies and whether NeuroFS performance remains stable for these different topologies.

To achieve this aim, we conduct two experiments. In the first experiment, we analyze the topology of five networks that are trained and initialized with different random seeds. In other words, they start with different sparse connectivities at initialization and have different training paths. In the second experiment, we analyze the topology of five networks initialized with the same sparse connectivity (using a similar random seed) and trained with different random seeds. For both experiments, we measure the topology distance among networks using a metric introduced in Liu et al. (2020), called NNSTD. It measures the distance of two sparse networks; NNSTD of 0 means that two networks are identical, and 1 means completely different.

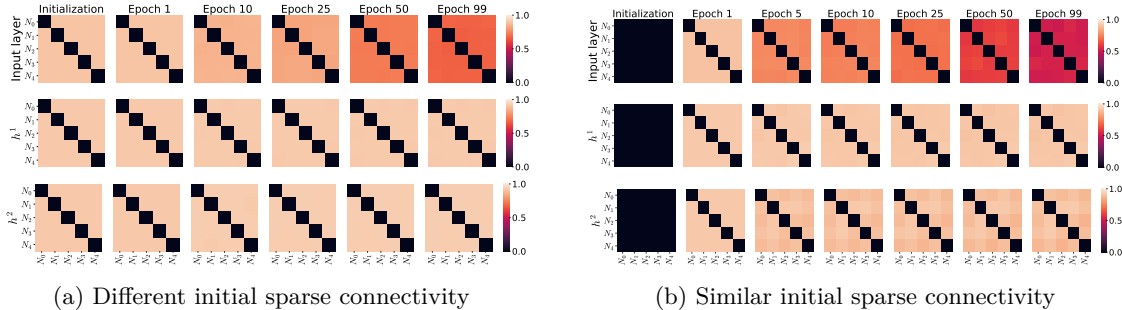

(a) Different initial sparse connectivity          (b) Similar initial sparse connectivity

Figure 5: Topology distance of five MLPs with (a) different and (b) similar initial sparse connectivity (topology). Input layers converge to relatively similar topologies in both cases, while hidden layers remain distant. $N_i$ refers to the network trained with $i^{th}$ random seed.

Table 3: NeuroFS Classification Accuracy (%) on the MNIST dataset for five networks ($K = 50$).

| | $N_0$ | $N_1$ | $N_2$ | $N_3$ | $N_4$ |
|---|---|---|---|---|---|
| NeuroFS (Different initial sparse connectivity) | 95.6 | 95.3 | 95.5 | 94.6 | 95.2 |
| NeuroFS (Similar initial sparse connectivity) | 95.6 | 94.4 | 96.2 | 95.4 | 95.8 |

We perform both experiments on the MNIST dataset to find the $K = 50$ most important features. The topology distance of the networks at different epochs are depicted in Figure 5 as 2d heatmaps. Each row depicts the distances for one layer of different networks. Each tile in the heatmaps refers to the distance between two layers of two networks. In these figures, $N_i$ refers to the network trained with $i^{th}$ random seed. The corresponding accuracies are shown in Table 3.

In Figure 5a, the networks are very distant at the beginning as their sparse connectivity (topology) initialized differently. During training, while their hidden layers remain distant, their input layers become more similar. Considering these figures and comparing them with the results in Table 3, it can be observed that while the feature selection remains almost the same, the network topologies do not. This indicates that NeuroFS can find several well-performing networks.

The similarity of the network topologies in Figure 5b almost match the pattern of Figure 5a. While the networks start from the same sparse connectivity, they become distant at the next epoch when they start training with different random seeds. This indicates that NeuroFS explores various connectivities during training. Interestingly, in the end, the converged input layers are more similar to each other than the experiment 1, due to the similar sparse connectivity at initialization. As shown in Table 3, the corresponding accuracies are close together. Experiment 2 confirms the observations in experiment 1, where NeuroFS finds distant topologies with very close feature selection performance.

To conclude, NeuroFS is robust to changes in topology. While it finds very different topologies overall, the input layers converge to relatively similar topologies, resulting in close feature selection performance.

## 5.2 Computational Efficiency of NeuroFS

In this section, we analyze the computational efficiency of NeuroFS. We present the number of training FLOPs and the number of parameters of NeuroFS and compare it with its neural network-based competitors.

Estimating the FLOPs (floating-point operations) and parameter count is a commonly used approach to analyze the efficiency gained by a sparse neural network compared to its dense equivalent network Evci et al. (2020); Sokar et al. (2021). *Number of parameters* indicates the size of the model, which directly affects the memory consumption and also computational complexity. *FLOPs* estimates the time complexity of an algorithm independently of its implementation. In addition, since existing deep learning hardware is not optimized for sparse matrix computations, most methods for obtaining sparse neural networks only simulate sparsity using a binary mask over the weights. Consequently, the running time of these methods does not reflect their efficiency. Besides, developing proper pure sparse implementations for sparse neural networks is currently a highly researched topic pursued by the community Hooker (2021). Thus, as our paper is, in its essence, theoretical, we decided to let this engineering research aspect for future work. Therefore, we also use parameter and FLOPs count to analyze efficiency.

To give an intuitive overview of the efficiency of NeuroFS, we compare NeuroFS with its neural network-based competitors. We compute the FLOPs and number of parameters of two dense MLPs with one ($Dense_1$) and three hidden layers ($Dense_3$). These are the architectures used by LassoNet and STG, respectively. However, it should be noted that LassoNet might require several rounds of training for the dense model. Therefore, we have also computed the actual training FLOPs for LassoNet. In addition, as the computational cost of QuickSelection is similar to our method, we refer to both NeuroFS and QuickSelection as *Sparse*.

As explained in Section 3.2.1, the sparsity/density level is determined by the $\varepsilon$. The density level of the network ($D$), the number of parameters and FLOP count of NeuroFS, and the compared methods are shown in Table 4. We estimate the FLOP count for the considered methods, using the implementation provided by Evci et al..

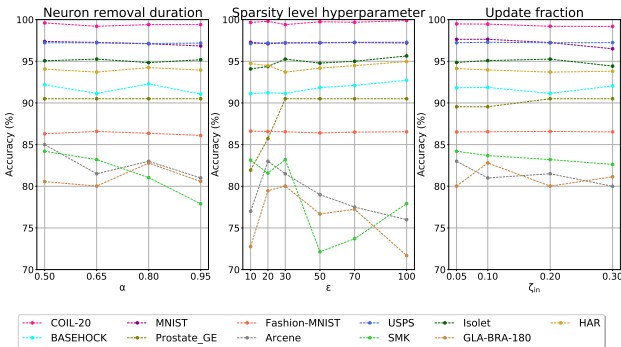

Figure 6: Effect of hyperparameters on the performance of the algorithm ($K = 100$).

As can be seen from Table 4, NeuroFS and QuickSelection (*Sparse*) have the least number of parameters and FLOPs among the considered architectures on all considered datasets, particularly on high-dimensional datasets. In addition, as discussed in Section 4.2, NeuroFS outperforms LassoNet, STG, and QuickSelection, in terms of accuracy on most cases considered. In short, NeuroFS is efficient in terms of memory requirements and computational costs while finding the most informative subset of the features on real-world benchmarks, including low and high-dimensional datasets.

## 5.3 Hyperparameters Effect

In this section, we analyze the effect of hyperparameters of NeuroFS on the quality of the selected features. The hyperparameters include neuron removal duration fraction $\alpha$, hyperparameter determining sparsity level $\varepsilon$, and the update fraction of the input layer $\zeta_{in}$. We try different sets of values for each of these hyperparameters and measure the performance of NeuroFS when selecting $K = 100$ features. The results are presented in Figure 6.

The results of most datasets are stable for different sets of hyperparameter values. However, high-dimensional datasets with few samples ($d \geq 10000$ and $m \leq 200$) are sensitive to the sparsity level hyperparameter. The feature selection performance decreases for higher densities; this might come from over-fitting of the network for large parameter count and a low number of training samples. We select $\alpha = 0.65$, $\varepsilon = 30$, and $\zeta_{in} = 0.2$ as the final values for the other experiments.

## 6 Conclusion

This paper proposes a novel supervised feature selection method named NeuroFS. NeuroFS introduces dynamic neuron evolution in the training process of a sparse neural network to find an informative set of features. By evaluating NeuroFS on real-world benchmark datasets, we demonstrated that it achieves the highest ranking-based score among the considered state-of-the-art supervised feature selection models. However, due to the general lack of knowledge on optimally implementing sparse neural networks during training, NeuroFS does not take full advantage of its theoretical high computational and memory advantages. We let the development of this challenging research direction for future work, hopefully, in a greater joint effort of the community.

Table 4: Number of parameters ($\times 10^5$) and Number of training FLOPs ($\times 10^{12}$) of NeuroFS (*Sparse*) and the equivalent dense MLPs on different datasets.

| Dataset | Density | #parameters ($\times 10^5$) | | | #FLOPs ($\times 10^{12}$) | | | |
|---|---|---|---|---|---|---|---|---|
| | | *Sparse* | $Dense_1$ | $Dense_3$ | *Sparse* | $Dense_1$ | $Dense_3$ | LassoNet |
| COIL-20 | 6.29% | 1.91 | 10.34 | 30.34 | 0.13 | 0.72 | 2.10 | 4.5 |
| MNIST | 6.57% | 1.84 | 7.94 | 27.94 | 6.66 | 28.60 | 100.64 | 371.0 |
| Fashion-MNIST | 6.57% | 1.84 | 7.94 | 27.94 | 6.66 | 28.60 | 100.64 | 439.8 |
| USPS | 7.40% | 1.68 | 2.66 | 22.66 | 0.76 | 1.19 | 10.12 | 10.9 |
| Isolet | 7.36% | 1.95 | 6.43 | 26.43 | 0.73 | 2.41 | 9.90 | 23.6 |
| HAR | 6.73% | 1.73 | 5.67 | 25.67 | 0.77 | 2.50 | 11.33 | 20.3 |
| BASEHOCK | 4.34% | 2.98 | 48.64 | 68.64 | 0.36 | 5.82 | 8.21 | 21.4 |
| Arcene | 3.77% | 4.52 | 100.02 | 120.02 | 0.05 | 1.20 | 1.44 | 5.8 |
| Prostate_GE | 4.15% | 3.31 | 59.68 | 79.68 | 0.02 | 0.37 | 0.49 | 1.9 |
| SMK-CAN-187 | 3.42% | 7.52 | 199.95 | 219.95 | 0.07 | 1.79 | 1.97 | 4.4 |
| GLA-BRA-180 | 3.18% | 16.29 | 491.55 | 511.55 | 0.18 | 5.31 | 5.52 | 12.6 |

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

# A    Ablation Study: Gradient vs Random Policy for Weight and Neuron Selection

This Appendix discusses the effect of gradient-based weights and neuron selection in NeuroFS by performing an ablation study. We use random growth instead of the gradient to measure the importance of weights and neurons. We call this method NeuroFS[w/oGradient]. The settings of this experiment is similar to Section 4.2. The results are presented in Figure 7.

In Figure 7, NeuroFS outperforms NeuroFS[w/oGradient] in most cases. While the results of these methods are relatively close on some datasets, on the Coil-20, SMK, GLA-BRA-180, and Arcene datasets, there is a large gap between the results. It can be concluded that NeuroFS performs more stable than NeuroFS[w/oGradient]. While random growth of weights and neurons might lead to better results in some cases, it can not ensure a stable performance across different datasets.

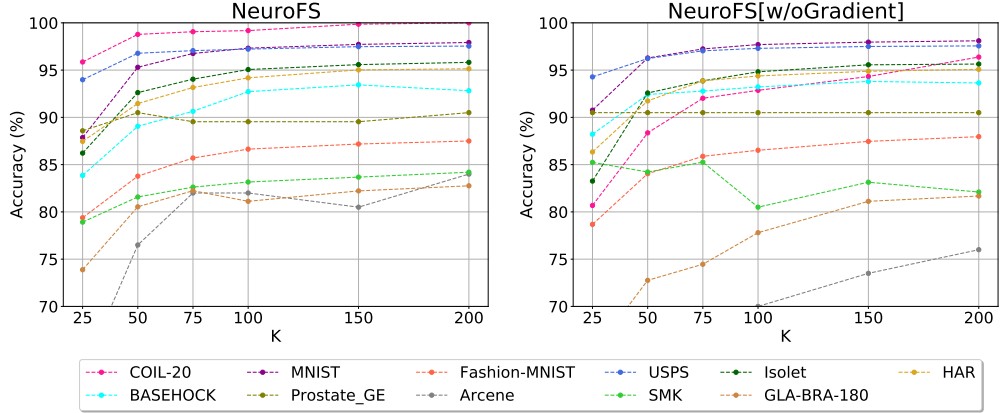

Figure 7: Gradient (left) vs. random (right) weight and neuron growth policy comparison.

# B    Comparison with HSICLasso-based Feature Selection Methods

In this section, we compare NeuroFS with two HSIC-based feature selection methods. We select two algorithms including, HSICLasso[8] Yamada et al. (2014) and HSICLassoVI[9] Koyama et al. (2022) and used the default hyperparameters used in the corresponding code repositories. The results are presented in Table 5. As can be seen in this table, NeuroFS outperforms these methods in seven cases while performing very close to the best performer in the other cases (less than a 1% difference in accuracy).

Table 5: Supervised feature selection comparison (average classification accuracy for $K$ values in [25, 50, 75, 100, 150, 200] (%)) with HSICLasso-based methods. Empty entries show that the corresponding experiments ran into error. Bold and italic fonts indicate the best and second-best performer, respectively.

| Method | Low-dimensional Datasets | | | | | | High-dimensional Datasets | | | | |
| | COIL-20 | MNIST | Fashion-MNIST | USPS | Isolet | HAR | BASEHOCK | Prostate_GE | Arcene | SMK | GLA-BRA-180 |
|---|---|---|---|---|---|---|---|---|---|---|---|
| Baseline | 100.0 | 97.92 | 88.3 | 97.58 | 96.03 | 95.05 | 91.98 | 80.95 | 77.5 | 86.84 | 72.22 |
| NeuroFS | *98.79±0.22* | **95.48 ± 0.34** | **85.03 ± 0.15** | **96.68 ± 0.16** | **93.22 ± 0.11** | **92.74 ± 0.23** | **90.42 ± 0.80** | *89.70±0.72* | **78.00 ± 1.78** | *82.36±0.98* | *80.46±0.99* |
| HSICLasso | **99.32 ± 0.00** | *93.80±0.00* | 82.73±0.00 | 96.03 ± 0.00 | *91.07±0.00* | *92.68±0.00* | *88.62±0.00* | **90.50 ± 0.00** | *77.50±0.00* | 70.60 ± 0.00 | **80.55 ± 0.00** |
| HSICLassoVI | 92.83 ± 0.00 | - | 75.72 ± 0.00 | *96.15±0.00* | 75.43 ± 0.00 | 89.10 ± 0.00 | - | - | - | **83.33 ± 0.00** | - |

# C    Supervised Feature Selection Comparison using Different Classifiers for Evaluation

To show that the evaluation results are not biased by the chosen classifier, we measure the classification accuracy by using two other widely-used classifiers including KNN[10] and ExtraTrees[11]. The classification

---

[8]https://github.com/riken-aip/pyHSICLasso

[9]https://github.com/nttcom/HSICLassoVI

[10]https://scikit-learn.org/stable/modules/generated/sklearn.neighbors.KNeighborsClassifier.html

[11]https://scikit-learn.org/stable/modules/generated/sklearn.ensemble.ExtraTreesClassifier.html

Table 6: Supervised feature selection comparison (classification accuracy for $K = 50$ (%)) using different classifiers. Empty entries show that the corresponding experiments exceeded the time limit (12 hours). Bold and italic fonts indicate the best and second-best performer, respectively.

| Method | Low-dimensional Datasets | | | | | | High-dimensional Datasets | | | | |
|---|---|---|---|---|---|---|---|---|---|---|---|
| | COIL-20 | MNIST | Fashion-MNIST | USPS | Isolet | HAR | BASEHOCK | Prostate_GE | Arcene | SMK | GLA-BRA-180 |
| SVM | | | | | | | | | | | |
| Baseline | 100.0 | 97.92 | 88.3 | 97.58 | 96.03 | 95.05 | 91.98 | 80.95 | 77.5 | 86.84 | 72.22 |
| NeuroFS | 98.78±0.29 | **95.30±0.41** | **83.78±0.64** | **96.78±0.17** | **92.62±0.40** | 91.46±0.72 | *89.06±2.46* | **90.50±0.00** | *76.50±2.55* | *81.58±1.68* | **80.54±4.96** |
| LassoNet | 97.16±1.06 | *94.46±0.21* | 82.58±0.10 | 95.94±0.15 | 84.90±0.22 | *93.74±0.39* | 87.18±0.58 | *88.58±2.35* | 71.00±2.00 | 80.52±2.69 | *74.46±4.78* |
| STG | **99.32±0.40** | 93.20±0.62 | 82.36±0.52 | *96.62±0.34* | 85.82±2.83 | 91.22±1.23 | 85.12±1.86 | 84.78±3.55 | 71.00±2.55 | 80.25±2.95 | 70.00±4.08 |
| QS | 96.52±1.53 | 93.62±0.49 | 80.82±0.51 | 95.52±0.27 | 89.78±1.80 | 91.96±1.04 | 87.22±1.22 | 76.20±7.53 | 74.38±4.80 | 80.90±2.20 | 72.20±2.80 |
| Fisher_score | 74.00±0.00 | 81.90±0.00 | 67.80±0.00 | 91.00±0.00 | 67.40±0.00 | 79.80±0.00 | **90.20±0.00** | **90.50±0.00** | 67.50±0.00 | 73.70±0.00 | 63.90±0.00 |
| CIFE | 59.40±0.00 | 89.30±0.00 | 66.90±0.00 | 61.30±0.00 | 59.80±0.00 | 84.20±0.00 | 77.40±0.00 | 47.60±0.00 | 52.50±0.00 | **81.60±0.00** | 58.30±0.00 |
| ICAP | *99.30±0.00* | 89.00±0.00 | 59.50±0.00 | 95.20±0.00 | 75.10±0.00 | 88.70±0.00 | **90.20±0.00** | 57.10±0.00 | 70.00±0.00 | 73.70±0.00 | 72.20±0.00 |
| RFS | 95.80±0.00 | - | - | 95.80±0.00 | *91.50±0.00* | **94.00±0.00** | 85.00±0.00 | **90.50±0.00** | **77.50±0.00** | - | - |
| KNN | | | | | | | | | | | |
| Baseline | 100.0 | 96.91 | 84.96 | 97.37 | 88.14 | 87.85 | 78.7 | 76.19 | 92.5 | 73.68 | 69.44 |
| NeuroFS | *99.80±0.28* | **91.64±0.57** | **80.12±0.87** | **96.18±0.49** | *85.96±1.53* | 84.64±1.77 | 87.14±2.69 | *85.86±4.67* | 74.00±5.15 | **78.97±4.55** | 64.42±5.38 |
| LassoNet | 98.84±0.20 | *91.38±0.36* | 79.30±0.20 | *95.70±0.26* | 79.22±0.47 | *88.70±0.57* | 88.96±1.20 | 82.86±3.80 | 67.50±7.75 | *74.74±6.34* | **68.90±4.07** |
| STG | **99.94±0.12** | 87.16±0.64 | 77.65±0.48 | 95.14±0.45 | 83.16±3.42 | 87.86±0.39 | 81.10±1.93 | 81.00±0.00 | *75.00±5.24* | 71.08±6.44 | 58.90±7.52 |
| QS | 98.80±0.38 | 89.30±0.76 | 76.65±0.51 | 95.17±0.45 | 82.38±3.12 | 85.88±1.13 | 86.02±0.97 | 65.47±8.37 | *75.00±3.54* | 69.08±2.87 | *66.70±0.00* |
| Fisher_score | 95.80±0.00 | 80.20±0.00 | 63.70±0.00 | 88.80±0.00 | 74.10±0.00 | 81.10±0.00 | *89.50±0.00* | 85.70±0.00 | 70.00±0.00 | 65.80±0.00 | 50.00±0.00 |
| CIFE | 71.20±0.00 | 82.90±0.00 | 61.60±0.00 | 59.60±0.00 | 44.60±0.00 | 71.80±0.00 | 68.40±0.00 | 57.10±0.00 | 70.00±0.00 | 71.10±0.00 | 44.40±0.00 |
| ICAP | 98.60±0.00 | 83.40±0.00 | 59.30±0.00 | 94.00±0.00 | 59.00±0.00 | 82.70±0.00 | **91.70±0.00** | 66.70±0.00 | 65.00±0.00 | 71.10±0.00 | 61.10±0.00 |
| RFS | 97.20±0.00 | - | - | 95.40±0.00 | **87.20±0.00** | **90.30±0.00** | 78.70±0.00 | **90.50±0.00** | **85.00±0.00** | - | - |
| ExtraTrees | | | | | | | | | | | |
| Baseline | 100.0 | 96.9 | 87.39 | 96.51 | 94.04 | 93.59 | 96.99 | 85.71 | 82.5 | 78.95 | 69.44 |
| NeuroFS | *99.94±0.12* | **93.68±0.43** | **84.26±0.55** | **95.44±0.27** | **91.46±0.73** | 85.48±1.46 | 89.96±1.89 | **90.50±0.00** | 75.00±5.24 | **81.75±4.16** | *75.46±6.71* |
| LassoNet | 99.76±0.12 | *92.96±0.15* | *83.68±0.13* | *94.86±0.22* | 84.94±0.62 | **91.12±0.30** | 92.08±0.36 | *89.54±1.92* | 73.50±4.64 | 77.88±6.77 | **76.12±3.80** |
| STG | **100.00±0.00** | 90.38±0.42 | 82.05±0.48 | 94.32±0.21 | 88.50±2.15 | 88.68±0.42 | 83.56±1.62 | 83.84±3.80 | *79.00±3.39* | 75.78±5.10 | 71.08±2.24 |
| QS | 99.25±0.47 | 91.95±0.58 | 81.28±0.54 | 94.28±0.40 | 88.78±1.86 | 87.86±0.72 | 86.80±0.91 | 77.38±5.19 | 73.75±4.15 | 75.00±1.30 | 75.00±0.00 |
| Fisher_score | 96.86±0.43 | 84.86±0.15 | 72.06±0.08 | 90.94±0.24 | 81.42±0.59 | 85.50±0.30 | *92.50±0.00* | **90.50±0.00** | 60.00±1.58 | 74.22±1.95 | 63.90±0.00 |
| CIFE | 74.70±0.00 | 87.60±0.00 | 68.40±0.00 | 82.70±0.00 | 55.40±0.00 | 85.30±0.00 | 85.20±0.00 | 52.40±0.00 | 50.00±0.00 | *81.60±0.00* | 69.40±0.00 |
| ICAP | 99.70±0.00 | 87.80±0.00 | 65.50±0.00 | 93.50±0.00 | 70.60±0.00 | 89.20±0.00 | **95.00±0.00** | 81.00±0.00 | **80.00±0.00** | 78.90±0.00 | 63.90±0.00 |
| RFS | 98.30±0.00 | - | - | 94.70±0.00 | *90.40±0.00* | *89.70±0.00* | 86.50±0.00 | **90.50±0.00** | 75.00±0.00 | - | - |

accuracy results are presented in Table 6 for $K = 50$. As can be seen in this table, the overall performance of all methods is consistent across different classifiers in most cases.

## D  Comparison to RigL

In this section, we compare NeuroFS with RigL Evci et al. (2020), which is a DST method mainly designed for classification; it uses gradient for weight regrowth when updating the sparse connectivity in the DST framework. To adapt RigL to perform feature selection, while trying to keep a fair comparison, we take the most straightforward approach: at the end of the training process with RigL, we use neuron strength (same as in NeuroFS) on the trained network to derive the indices of the important features. We train a 3-layer MLP with RigL for 100 epochs. RigL training algorithm updates the sparse connectivity at each epoch by removing $\zeta_h$ of the weights with the lowest magnitude and adding the same number of connections as the dropped ones to the network, among the non-existing connections with the highest gradient magnitude. When the training is finished we select the top $K$ features corresponding to the neurons with the highest neuron strength as the selected features. The main difference between NeuroFS and feature selection using RigL is the input layer neuron removal and addition. While RigL updates only the sparse connectivity in all layers, NeuroFS updates also the neurons in the input layer to gradually decrease the number of active neurons (neurons with at least one non-zero connection) to be suited for feature selection. All the experimental settings are similar to NeuroFS (Section 4.2), such as $\epsilon = 30$, $\zeta_h = 0.2$, training epochs, activation functions, batch size, learning rate, and etc. We measure the performance of feature selection using RigL for several values of $K \in \{25, 50, 75, 100, 150, 200\}$. The results are presented in Table 7 and Figure 8.

As can be seen in Table 7, NeuroFS outperforms feature selection with RigL in most cases in terms of classification accuracy. On low-dimensional datasets, RigL performs closely to NeuroFS and even outperforms it in some cases, particularly for large values of $K$; while for small values of $K$, e.g., 25 or 50, the performance gap is larger than the larger $K$ values. On the other hand on high-dimensional datasets, NeuroFS outperforms RigL with a large gap in most datasets considered except SMK where they perform very closely. It can be concluded that the performance gap is usually high in cases where the proportion of selected features to the total number of features is low, e.g., selecting a low number of features in low-dimensional datasets and feature selection from high-dimensional datasets. Relatively similar behavior exists in feature selection using QuickSelection, particularly in high-dimensional datasets such as BASEHOCK, Prostate_GE, and GLA-BRA-180 (See Table 8). The reason behind this is that when the search space becomes large (all input

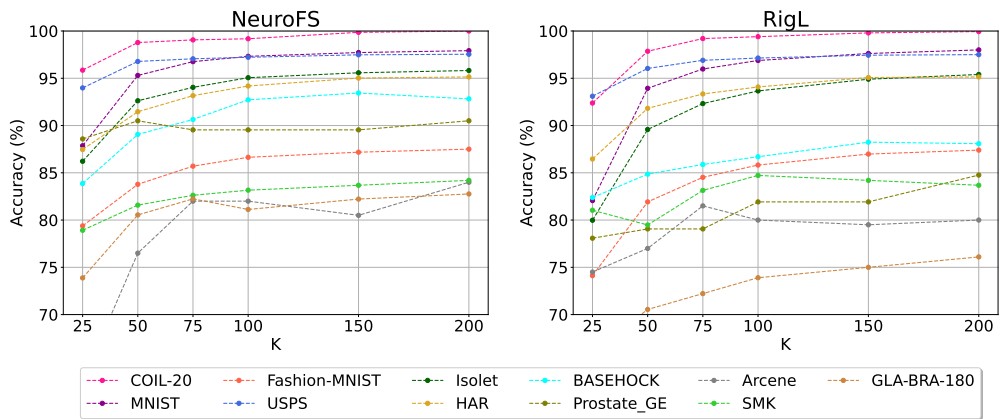

Figure 8: NeuroFS (left) vs. feature selection using RigL (right) comparison.

features), finding a low fraction of informative features becomes difficult for QuickSelection and RigL and the neuron strength might not be informative on its own. Therefore, NeuroFS reduces the search space by removing uninformative features during training, thus allowing the high-magnitude weights to be assigned to a limited set of the most informative features. This indicates the importance of the neuron removal scheme in NeuroFS.

Table 7: Supervised feature selection comparison (classification accuracy (%)) with RigL for $K \in \{25, 50, 75, 100, 150, 200\}$. Bold fonts indicate the best performer for each dataset.

| Dataset | Method | $K = 25$ | $K = 50$ | $K = 75$ | $K = 100$ | $K = 150$ | $K = 200$ |
|---|---|---|---|---|---|---|---|
| COIL-20 | NeuroFS | **95.86 ± 1.31** | **98.78 ± 0.29** | 99.06 ± 0.12 | 99.18 ± 0.5 | **99.86 ± 0.28** | **100.0 ± 0.0** |
| | RigL | 92.38 ± 3.2 | 97.86 ± 1.32 | **99.2 ± 0.43** | **99.4 ± 0.43** | 99.8 ± 0.28 | 99.94 ± 0.12 |
| MNIST | NeuroFS | **87.86 ± 1.77** | **95.3 ± 0.41** | **96.76 ± 0.22** | **97.32 ± 0.17** | **97.72 ± 0.1** | 97.92 ± 0.07 |
| | RigL | 82.06 ± 0.99 | 93.94 ± 0.63 | 95.98 ± 0.51 | 96.88 ± 0.22 | 97.62 ± 0.07 | **98.0 ± 0.09** |
| Fashion-MNIST | NeuroFS | **79.38 ± 0.96** | **83.78 ± 0.64** | **85.7 ± 0.28** | **86.64 ± 0.21** | **87.18 ± 0.16** | **87.5 ± 0.17** |
| | RigL | 74.12 ± 1.59 | 81.92 ± 0.87 | 84.52 ± 0.72 | 85.82 ± 0.23 | 86.98 ± 0.12 | 87.4 ± 0.23 |
| USPS | NeuroFS | **93.98 ± 0.87** | **96.78 ± 0.17** | **97.06 ± 0.15** | **97.22 ± 0.12** | **97.48 ± 0.04** | **97.54 ± 0.1** |
| | RigL | 93.1 ± 0.62 | 96.04 ± 0.58 | 96.9 ± 0.24 | 97.14 ± 0.1 | 97.44 ± 0.1 | 97.5 ± 0.11 |
| Isolet | NeuroFS | **86.22 ± 0.84** | **92.62 ± 0.4** | **94.04 ± 0.34** | **95.06 ± 0.31** | **95.58 ± 0.29** | **95.82 ± 0.31** |
| | RigL | 79.98 ± 2.25 | 89.58 ± 1.24 | 92.32 ± 0.56 | 93.66 ± 0.58 | 94.9 ± 0.56 | 95.4 ± 0.32 |
| HAR | NeuroFS | **87.46 ± 0.79** | 91.46 ± 0.72 | 93.16 ± 0.79 | **94.18 ± 0.29** | **95.02 ± 0.35** | **95.14 ± 0.21** |
| | RigL | 86.46 ± 1.47 | **91.82 ± 0.3** | **93.34 ± 0.47** | 94.08 ± 0.26 | **95.08 ± 0.26** | **95.14 ± 0.37** |
| BASEHOCK | NeuroFS | **83.86 ± 3.38** | **89.06 ± 2.46** | **90.64 ± 2.35** | **92.72 ± 1.5** | **93.44 ± 0.91** | **92.82 ± 1.13** |
| | RigL | 82.38 ± 2.85 | 84.86 ± 3.04 | 85.88 ± 2.32 | 86.7 ± 1.54 | 88.24 ± 1.7 | 88.08 ± 1.92 |
| Prostate_GE | NeuroFS | **88.58 ± 2.35** | **90.5 ± 0.0** | **89.54 ± 1.92** | **89.54 ± 1.92** | **89.54 ± 1.92** | **90.5 ± 0.0** |
| | RigL | 78.08 ± 6.46 | 79.06 ± 7.11 | 79.06 ± 8.83 | 81.92 ± 8.18 | 81.92 ± 6.33 | 84.76 ± 1.88 |
| Arcene | NeuroFS | 63.0 ± 4.85 | 76.5 ± 2.55 | **82.0 ± 4.0** | **82.0 ± 1.87** | **80.5 ± 4.3** | **84.0 ± 3.39** |
| | RigL | **74.5 ± 4.3** | **77.0 ± 3.32** | 81.5 ± 4.64 | 80.0 ± 4.47 | 79.5 ± 4.3 | 80.0 ± 4.18 |
| SMK | NeuroFS | 78.92 ± 1.68 | **81.58 ± 1.68** | 82.62 ± 2.12 | 83.16 ± 1.27 | 83.68 ± 1.04 | **84.2 ± 0.0** |
| | RigL | **81.04 ± 1.99** | 79.48 ± 4.81 | **83.14 ± 3.15** | **84.72 ± 1.95** | **84.2 ± 3.73** | 83.68 ± 2.55 |
| GLA-BRA-180 | NeuroFS | **73.88 ± 3.8** | **80.54 ± 4.96** | **82.24 ± 3.31** | **81.12 ± 2.05** | **82.22 ± 1.32** | **82.76 ± 2.71** |
| | RigL | 66.1 ± 3.22 | 70.54 ± 4.16 | 72.22 ± 4.98 | 73.9 ± 3.76 | 75.0 ± 3.07 | 76.1 ± 4.16 |

# E  Comparison Results

The detailed results for each K value are presented in Table 8.

Table 8: Supervised feature selection comparison (classification accuracy for various $K$ values (%)). Empty entries show that the corresponding experiments exceeded the time limit (12 hours). Bold and italic fonts indicate the best and second-best performer, respectively.

### (a) K = 25

| Method | Low-dimensional Datasets | | | | | | High-dimensional Datasets | | | | |
| --- | --- | --- | --- | --- | --- | --- | --- | --- | --- | --- | --- |
| | COIL-20 | MNIST | Fashion-MNIST | USPS | Isolet | HAR | BASEHOCK | Prostate_GE | Arcene | SMK | GLA-BRA-180 |
| Baseline | 100.0 | 97.92 | 88.3 | 97.58 | 96.03 | 95.05 | 91.98 | 80.95 | 77.5 | 86.84 | 72.22 |
| NeuroFS | *95.86±1.31* | **87.86 ± 1.77** | **79.38 ± 0.96** | 93.98±0.87 | **86.22 ± 0.84** | 87.46 ± 0.79 | 83.86 ± 3.38 | *88.58±2.35* | 63.00 ± 4.85 | *78.92±1.68* | *73.88±3.80* |
| LassoNet | 92.72 ± 0.85 | *86.40±1.26* | *78.68±0.55* | *94.04±0.38* | 76.48 ± 0.39 | **93.00 ± 0.31** | 84.48 ± 0.86 | *88.58±2.35* | 69.00 ± 2.55 | 76.84 ± 5.34 | **76.12 ± 4.19** |
| STG | **97.02 ± 1.41** | 85.24 ± 1.89 | 77.44 ± 0.53 | 94.04±0.46 | 77.16±4.34 | 87.48±0.80 | 82.38 ± 1.36 | 85.72 ± 3.00 | 69.00 ± 5.15 | 77.38 ± 3.57 | 67.22 ± 4.78 |
| QS | 91.00 ± 4.21 | 85.25 ± 1.47 | 71.57 ± 1.97 | 93.00 ± 0.81 | 72.56 ± 6.53 | 87.14 ± 1.74 | 83.80 ± 1.61 | 71.43 ± 12.16 | *73.75±8.20* | 76.97 ± 7.52 | 69.45 ± 2.75 |
| Fisher_score | 24.70 ± 0.00 | 74.40 ± 0.00 | 53.10 ± 0.00 | 82.00 ± 0.00 | 57.40 ± 0.00 | 77.10 ± 0.00 | *85.50±0.00* | 65.00 ± 0.00 | 68.40 ± 0.00 | | 58.30 ± 0.00 |
| CIFE | 50.70 ± 0.00 | 80.90 ± 0.00 | 63.40 ± 0.00 | 50.20 ± 0.00 | 56.00 ± 0.00 | 80.20 ± 0.00 | 76.20 ± 0.00 | 61.90 ± 0.00 | 67.50 ± 0.00 | **81.60 ± 0.00** | 61.10 ± 0.00 |
| ICAP | 94.40 ± 0.00 | 81.60 ± 0.00 | 50.10 ± 0.00 | 89.90 ± 0.00 | 67.10 ± 0.00 | 84.50 ± 0.00 | **89.20 ± 0.00** | 47.60 ± 0.00 | **77.50 ± 0.00** | 78.90 ± 0.00 | 69.40 ± 0.00 |
| RFS | 88.20 ± 0.00 | - | - | **94.80 ± 0.00** | 76.50±0.00 | *88.90±0.00* | 80.20 ± 0.00 | **90.50 ± 0.00** | **77.50 ± 0.00** | - | - |

### (b) K = 50

| Method | Low-dimensional Datasets | | | | | | High-dimensional Datasets | | | | |
| --- | --- | --- | --- | --- | --- | --- | --- | --- | --- | --- | --- |
| | COIL-20 | MNIST | Fashion-MNIST | USPS | Isolet | HAR | BASEHOCK | Prostate_GE | Arcene | SMK | GLA-BRA-180 |
| Baseline | 100.0 | 97.92 | 88.3 | 97.58 | 96.03 | 95.05 | 91.98 | 80.95 | 77.5 | 86.84 | 72.22 |
| NeuroFS | 98.78 ± 0.29 | **95.30 ± 0.41** | **83.78 ± 0.64** | **96.78 ± 0.17** | **92.62 ± 0.40** | 91.46 ± 0.72 | *89.06±2.46* | **90.50 ± 0.00** | 76.50±2.55 | *81.58±1.68* | **80.54 ± 4.96** |
| LassoNet | 97.16 ± 1.06 | *94.46±0.21* | *82.58±0.10* | 95.94 ± 0.15 | 84.90 ± 0.22 | *93.74±0.39* | 87.18 ± 0.58 | 88.58±2.35 | 71.00 ± 2.00 | 80.52 ± 2.69 | *74.46±4.78* |
| STG | **99.32 ± 0.40** | 93.20 ± 0.62 | 82.36 ± 0.28 | *96.62±0.34* | 85.82 ± 2.83 | 91.22 ± 1.23 | 85.12 ± 1.86 | 84.78 ± 3.55 | 71.00 ± 2.55 | 80.25 ± 2.95 | 70.00 ± 4.08 |
| QS | 96.52 ± 1.53 | 93.62 ± 0.49 | 80.82 ± 0.51 | 95.52 ± 0.27 | 89.78 ± 1.80 | 91.96 ± 1.04 | 87.22 ± 1.22 | 76.20 ± 7.53 | 74.38 ± 4.80 | 80.90 ± 2.20 | 72.20 ± 2.80 |
| Fisher_score | 74.00 ± 0.00 | 81.90 ± 0.00 | 67.80 ± 0.00 | 91.00 ± 0.00 | 67.40 ± 0.00 | 79.80 ± 0.00 | 90.20 ± 0.00 | **90.50 ± 0.00** | 67.50 ± 0.00 | 73.70 ± 0.00 | 63.90 ± 0.00 |
| CIFE | 59.40 ± 0.00 | 89.30 ± 0.00 | 66.90 ± 0.00 | 61.30 ± 0.00 | 59.80 ± 0.00 | 84.20 ± 0.00 | 77.40 ± 0.00 | 47.60 ± 0.00 | 52.50 ± 0.00 | **81.60 ± 0.00** | 58.30 ± 0.00 |
| ICAP | *99.30±0.00* | 89.00 ± 0.00 | 59.50 ± 0.00 | 95.20 ± 0.00 | 75.10 ± 0.00 | 88.70 ± 0.00 | 90.20 ± 0.00 | 57.10 ± 0.00 | 70.00 ± 0.00 | 73.70 ± 0.00 | 72.20 ± 0.00 |
| RFS | 95.80 ± 0.00 | - | - | 95.80 ± 0.00 | *91.50±0.00* | **94.00 ± 0.00** | 85.00 ± 0.00 | **90.50 ± 0.00** | 77.50 ± 0.00 | - | - |

### (c) K = 75

| Method | Low-dimensional Datasets | | | | | | High-dimensional Datasets | | | | |
| --- | --- | --- | --- | --- | --- | --- | --- | --- | --- | --- | --- |
| | COIL-20 | MNIST | Fashion-MNIST | USPS | Isolet | HAR | BASEHOCK | Prostate_GE | Arcene | SMK | GLA-BRA-180 |
| Baseline | 100.0 | 97.92 | 88.3 | 97.58 | 96.03 | 95.05 | 91.98 | 80.95 | 77.5 | 86.84 | 72.22 |
| NeuroFS | 99.06 ± 0.12 | **96.76 ± 0.22** | **85.70 ± 0.28** | *97.06±0.15* | **94.04 ± 0.34** | 93.16 ± 0.79 | *90.64±2.35* | *89.54±1.92* | **82.00 ± 4.00** | **82.62 ± 2.12** | **82.24 ± 3.31** |
| LassoNet | 99.46 ± 0.35 | *96.00±0.09* | 83.92 ± 0.13 | 96.36 ± 0.08 | 91.00 ± 0.62 | *94.62±0.17* | 90.52 ± 0.27 | **90.50 ± 0.00** | 70.50 ± 2.45 | 78.94 ± 3.72 | *76.64±5.44* |
| STG | *99.68±0.22* | 95.52 ± 0.22 | *84.14±0.43* | 96.88 ± 0.23 | 90.10 ± 2.17 | 92.42 ± 1.11 | 85.52 ± 1.22 | 84.78 ± 3.55 | 71.00 ± 2.74 | 81.04 ± 4.21 | 71.08 ± 1.37 |
| QS | 98.17 ± 1.16 | 95.98 ± 0.33 | 83.80 ± 0.53 | 96.85 ± 0.05 | 93.04 ± 0.46 | 93.50 ± 0.77 | 87.55 ± 1.30 | 72.62 ± 9.78 | 76.88 ± 2.72 | *82.22±2.86* | 73.60 ± 1.40 |
| Fisher_score | 76.00 ± 0.00 | 87.10 ± 0.00 | 74.30 ± 0.00 | 94.40 ± 0.00 | 76.00 ± 0.00 | 81.70 ± 0.00 | 89.00 ± 0.00 | **90.50 ± 0.00** | 70.00 ± 0.00 | 76.30 ± 0.00 | 66.70 ± 0.00 |
| CIFE | 63.20 ± 0.00 | 92.70 ± 0.00 | 67.70 ± 0.00 | 68.00 ± 0.00 | 74.30 ± 0.00 | 84.80 ± 0.00 | 74.70 ± 0.00 | 47.60 ± 0.00 | 72.50 ± 0.00 | 76.30 ± 0.00 | 58.30 ± 0.00 |
| ICAP | 99.00 ± 0.00 | 92.40 ± 0.00 | 67.20 ± 0.00 | 95.30 ± 0.00 | 79.70 ± 0.00 | 89.20 ± 0.00 | **93.50 ± 0.00** | 57.10 ± 0.00 | 72.50 ± 0.00 | 71.10 ± 0.00 | 72.20 ± 0.00 |
| RFS | **99.70 ± 0.00** | - | - | **97.20 ± 0.00** | *93.90±0.00* | **94.90 ± 0.00** | 86.50 ± 0.00 | **90.50 ± 0.00** | *80.00±0.00* | - | - |

### (d) K = 100

| Method | Low-dimensional Datasets | | | | | | High-dimensional Datasets | | | | |
| --- | --- | --- | --- | --- | --- | --- | --- | --- | --- | --- | --- |
| | COIL-20 | MNIST | Fashion-MNIST | USPS | Isolet | HAR | BASEHOCK | Prostate_GE | Arcene | SMK | GLA-BRA-180 |
| Baseline | 100.0 | 97.92 | 88.3 | 97.58 | 96.03 | 95.05 | 91.98 | 80.95 | 77.5 | 86.84 | 72.22 |
| NeuroFS | 99.18 ± 0.50 | **97.32 ± 0.17** | **86.64 ± 0.21** | *97.22±0.12* | **95.06 ± 0.31** | 94.18 ± 0.29 | *92.72±1.50* | *89.54±1.92* | 82.00±1.87 | *83.16±1.27* | **81.12 ± 2.05** |
| LassoNet | 99.30 ± 0.00 | 96.64 ± 0.14 | 84.98 ± 0.18 | 97.04 ± 0.12 | 93.18 ± 0.22 | *95.14±0.29* | 90.96 ± 1.36 | **90.50 ± 0.00** | 72.00 ± 4.30 | 78.42 ± 4.20 | *79.46±2.83* |
| STG | *99.76±0.12* | 96.38 ± 0.35 | 85.20 ± 0.58 | 97.08 ± 0.18 | 92.64 ± 0.56 | 92.82 ± 0.74 | 85.96 ± 1.24 | 85.72 ± 3.00 | 75.50 ± 3.67 | 82.08 ± 3.87 | 72.20 ± 3.07 |
| QS | 98.28 ± 1.15 | *96.85±0.09* | *85.52±0.15* | 97.00 ± 0.14 | 94.22 ± 0.28 | 94.06 ± 0.48 | 89.02 ± 1.26 | 78.58 ± 9.82 | 78.12 ± 1.08 | **84.85 ± 2.16** | 73.60 ± 1.40 |
| Fisher_score | 80.20 ± 0.00 | 90.70 ± 0.00 | 79.60 ± 0.00 | 96.50 ± 0.00 | 79.80 ± 0.00 | 83.80 ± 0.00 | 89.70 ± 0.00 | **90.50 ± 0.00** | 65.00 ± 0.00 | 78.90 ± 0.00 | 66.70 ± 0.00 |
| CIFE | 67.70 ± 0.00 | 95.10 ± 0.00 | 69.20 ± 0.00 | 78.00 ± 0.00 | 81.20 ± 0.00 | 85.30 ± 0.00 | 74.40 ± 0.00 | 71.40 ± 0.00 | 65.00 ± 0.00 | 81.60 ± 0.00 | 58.30 ± 0.00 |
| ICAP | **100.00 ± 0.00** | 95.00 ± 0.00 | 77.70 ± 0.00 | 95.40 ± 0.00 | 82.80 ± 0.00 | 92.10 ± 0.00 | **94.00 ± 0.00** | 52.40 ± 0.00 | **82.50 ± 0.00** | 76.30 ± 0.00 | 69.40 ± 0.00 |
| RFS | **100.00 ± 0.00** | - | - | **97.40 ± 0.00** | *94.40±0.00* | **95.40 ± 0.00** | 86.70 ± 0.00 | **90.50 ± 0.00** | 80.00 ± 0.00 | - | - |

### (e) K = 150

| Method | Low-dimensional Datasets | | | | | | High-dimensional Datasets | | | | |
| --- | --- | --- | --- | --- | --- | --- | --- | --- | --- | --- | --- |
| | COIL-20 | MNIST | Fashion-MNIST | USPS | Isolet | HAR | BASEHOCK | Prostate_GE | Arcene | SMK | GLA-BRA-180 |
| Baseline | 100.0 | 97.92 | 88.3 | 97.58 | 96.03 | 95.05 | 91.98 | 80.95 | 77.5 | 86.84 | 72.22 |
| NeuroFS | 99.86 ± 0.28 | **97.72 ± 0.10** | **87.18 ± 0.16** | **97.48 ± 0.04** | *95.58±0.29* | 95.02 ± 0.35 | **93.44 ± 0.91** | *89.54±2.29* | **80.50 ± 4.30** | *83.68±1.04* | **82.22 ± 1.32** |
| LassoNet | 99.54 ± 0.20 | 97.42 ± 0.07 | 86.02 ± 0.15 | **97.48 ± 0.07** | 94.96 ± 0.15 | **95.72 ± 0.20** | 92.58 ± 0.62 | **90.50 ± 0.00** | 72.00 ± 1.87 | 78.38 ± 1.04 | *79.48±1.37* |
| STG | **100.00 ± 0.00** | 97.14 ± 0.08 | 86.28 ± 0.35 | 97.28 ± 0.12 | 94.20 ± 0.35 | 93.56 ± 0.59 | 86.42 ± 1.74 | 84.76 ± 4.67 | 75.00 ± 3.16 | 82.60 ± 4.27 | 72.76 ± 3.27 |
| QS | *99.92±0.13* | *97.70±0.12* | *86.88±0.32* | *97.42±0.11* | 95.48 ± 0.32 | 94.80 ± 0.24 | 89.80 ± 0.83 | 79.75 ± 6.19 | 78.75 ± 1.25 | **84.22 ± 3.23** | 75.00 ± 0.00 |
| Fisher_score | 81.20 ± 0.00 | 93.10 ± 0.00 | 83.60 ± 0.00 | 97.30 ± 0.00 | 83.00 ± 0.00 | 84.40 ± 0.00 | 91.70 ± 0.00 | **90.50 ± 0.00** | 65.00 ± 0.00 | 78.90 ± 0.00 | 63.90 ± 0.00 |
| CIFE | 71.90 ± 0.00 | 96.80 ± 0.00 | 75.60 ± 0.00 | 89.60 ± 0.00 | 85.70 ± 0.00 | 85.90 ± 0.00 | 79.20 ± 0.00 | 76.20 ± 0.00 | 55.00 ± 0.00 | 81.60 ± 0.00 | 72.20 ± 0.00 |
| ICAP | **100.00 ± 0.00** | 96.40 ± 0.00 | 81.70 ± 0.00 | 95.80 ± 0.00 | 89.30 ± 0.00 | 93.40 ± 0.00 | *93.20±0.00* | 42.90 ± 0.00 | 77.50 ± 0.00 | 71.10 ± 0.00 | 69.40 ± 0.00 |
| RFS | **100.00 ± 0.00** | - | - | 97.40 ± 0.00 | **95.90 ± 0.00** | *95.50±0.00* | 88.20 ± 0.00 | **90.50 ± 0.00** | *80.00±0.00* | - | - |

### (f) K = 200

| Method | Low-dimensional Datasets | | | | | | High-dimensional Datasets | | | | |
| --- | --- | --- | --- | --- | --- | --- | --- | --- | --- | --- | --- |
| | COIL-20 | MNIST | Fashion-MNIST | USPS | Isolet | HAR | BASEHOCK | Prostate_GE | Arcene | SMK | GLA-BRA-180 |
| Baseline | 100.0 | 97.92 | 88.3 | 97.58 | 96.03 | 95.05 | 91.98 | 80.95 | 77.5 | 86.84 | 72.22 |
| NeuroFS | **100.00 ± 0.00** | *97.92±0.07* | *87.50±0.17* | *97.54±0.10* | *95.82±0.31* | 95.14 ± 0.21 | 92.82 ± 1.13 | **90.50 ± 0.00** | **84.00 ± 3.39** | **84.20 ± 0.00** | **82.76 ± 2.71** |
| LassoNet | **100.00 ± 0.00** | 97.90 ± 0.00 | 86.66 ± 0.14 | **97.58 ± 0.07** | 95.34 ± 0.05 | *95.58±0.12* | *92.92±1.06* | **90.50 ± 0.00** | 73.50 ± 2.55 | 78.92 ± 1.68 | *80.02±2.06* |
| STG | **100.00 ± 0.00** | 97.46 ± 0.20 | 87.00 ± 0.26 | 97.38 ± 0.04 | 94.88 ± 0.07 | 93.70 ± 0.66 | 87.46 ± 1.03 | *86.68±3.56* | 77.00 ± 3.32 | 81.54 ± 3.34 | 73.88 ± 3.80 |
| QS | *99.50±0.53* | **98.00 ± 0.07** | **87.52 ± 0.15** | 97.50 ± 0.00 | **96.14 ± 0.08** | 94.76 ± 0.29 | 90.18 ± 0.66 | 79.75 ± 6.19 | *80.62±3.25* | 82.88±5.44 | 73.60 ± 1.40 |
| Fisher_score | 84.00 ± 0.00 | 94.50 ± 0.00 | 84.70 ± 0.00 | 97.50 ± 0.00 | 89.90 ± 0.00 | 85.80 ± 0.00 | 92.20 ± 0.00 | **90.50 ± 0.00** | 65.00 ± 0.00 | 78.90 ± 0.00 | 61.10 ± 0.00 |
| CIFE | 72.20 ± 0.00 | 97.60 ± 0.00 | 78.80 ± 0.00 | 96.30 ± 0.00 | 87.90 ± 0.00 | 85.90 ± 0.00 | 79.20 ± 0.00 | 76.20 ± 0.00 | 57.50 ± 0.00 | 78.90 ± 0.00 | 72.20 ± 0.00 |
| ICAP | 99.30 ± 0.00 | 97.60 ± 0.00 | 84.50 ± 0.00 | 96.90 ± 0.00 | 90.30 ± 0.00 | 93.30 ± 0.00 | **93.70 ± 0.00** | 42.90 ± 0.00 | 80.00 ± 0.00 | 76.30 ± 0.00 | 72.20 ± 0.00 |
| RFS | **100.00 ± 0.00** | - | - | 97.50 ± 0.00 | 95.70 ± 0.00 | **95.80 ± 0.00** | 89.00 ± 0.00 | **90.50 ± 0.00** | 80.00 ± 0.00 | - | - |

