# OpenReview forum: "Supervised Feature Selection with Neuron Evolution in Sparse Neural Networks"
_TMLR — Accepted by TMLR_

### Review · Reviewer_Cn4v · 2022-10-23

**Summary Of Contributions:**

The authors address a challenging and important problem of feature selection. The new approach is an embedded-type method that relies on ideas from sparse neural networks to train the model for prediction and FS. The main advantage of the approach is that it does not require training a dense network but rather sparsifies and regrows neurons to learn the prediction model while simultaneously identifying the best subset of features. They demonstrate the effectiveness of the approach using 11 real-world datasets.

**Audience:**

Yes

**Broader Impact Concerns:**

No ethical concerns

**Claims And Evidence:**

No

**Requested Changes:**

Here, I point out some suggestions to improve the paper:

The second sentence in the abstract is very long; try to rephrase it.

Unimportant-> this is a weird term; maybe you mean uninformative?
In the intro, you claim embedded methods are more efficient than other types, but then you evaluate the method as a wrapper. Furthermore, some filter methods are highly efficient, scaling linearly with the data, so I recommend relaxing your statement.

P2 “the proposed method, QuikSelection” this is confusing, quickselection is not proposed here

Equation 1, k should be capitalized

For unsupervised FS methods, it is worthwhile to cite
[1] Abid et al.  Concrete Autoencoders for Differentiable Feature Selection and Reconstruction
[2] Shaham et al. Deep Unsupervised Feature Selection by Discarding Nuisance and Correlated Features

Section 3.2, please add more details about the figure. Also, please move it to the same/adjacent page.
Section 3.2 more detail-? Details

Hyperparameters- Your use a 3-layer sparse MLP with 1000 neurons. Is this the architecture for all NN baselines? This is not clear

It is not clear how the parameters are optimized for all methods, do you have a validation set?

In table 2, what is the number of features selected by each method? This is crucial for understanding each method. If this is an average overall value of K it is really not informative.

You train SVM; what type? Kernel SVM? How is it tuned?

P12, STG, and LassoNet are not really dense NNs, since they sparsify the first layer during training; the number of flops they use varies and needs to be evaluated empirically by running the method on each data. This means that your statement is misleading, and the results of STG and LASSONet should be evaluated and added to table 4 as another column.



**Strengths And Weaknesses:**

Strengths: The problem is well-studied but still important for the community. The English level is satisfactory, and overall I enjoyed reading the paper. The problem statement and solutions are clear. The evaluation seems clear, and they further provide an evaluation of the sensitivity of the approach to hyperparameters and stability to different initializations. Related work is mostly well-written. The method is simple and leads to performance improvements while reducing flops.

Weakness: The method is embedded but is evaluated as a wrapper. Namely, the standard way to evaluate an embedded method is to use the prediction learned by the classifier/regressor that is learned in an end-to-end fashion. Instead, the authors train an additional classifier after their model selects features. While I understand the intuition behind this procedure, it is not the correct procedure to evaluate the capabilities of an embedded approach. Since they already put substantial effort into this procedure, I would recommend adding at least one or two evaluations in this suggested setting and comparing their model to other embedded approaches like LASSO, LASSO net, and STG… Furthermore, they haven’t presented any example that demonstrates that the model can accurately recover informative features in a noisy environment. This could be done with synthetic data or semi-synthetic like MADELLON or GISSETTE.

---

> ### Author Response · Authors · 2022-11-28
> **Response to Reviewer Cn4v (1/4)**
>
> We want to thank Reviewer Cn4v for their effort and time in reading and reviewing our paper. We thank reviewer Cn4v for pointing out the **importance of the studied problem** and the strengths of our work (**clear writing, satisfactory English level, clear evaluation, well-written related work, performance improvement with FLOPs reduction**). Please find the response to the comments below. We have incorporated the responses in the updated manuscript. We believe that these insightful suggestions have resulted in a significantly improved manuscript. If any responses are unclear or you wish additional changes, please let us know.
>
>
> **1. [...comparing their model to other embedded approaches…]**
>
> We thank the reviewer for raising this comment. Below, we first clarify the reason behind choosing this evaluation approach (1.1). Then, we report the network classification accuracy and discuss the architecture learned by each method (1.2). Finally, we report the classification accuracy for each method using classifiers other than the SVM classifier (1.3).
>
> **1.1. [Evaluation method]** We used the general evaluation approach for supervised feature selection methods, as suggested in [1] and widely used in the feature selection literature, e.g., [2, 4]. This evaluation approach aims to assess the quality of the selected features, which is the output of each algorithm. In this approach, we use the classification accuracy of a classifier (e.g., SVM classifier) trained on the subset of the features selected by each method for comparison. The higher the classification accuracy, the better the selected features are [1]. In this way, we can have a fair comparison among various (filter, wrapper, embedded) feature selection algorithms as the evaluation metric is not biased on the training method used by each algorithm.
>
> **1.2. [Network Accuracy]** Based on the reviewer’s suggestion, to evaluate the trained network, we measure the accuracy of the network trained within NeuoFS and compare it with the one of LassoNet and STG in the below table (in the following comment).
>
> Before discussing the results, we give some details about the training process of these three algorithms. NeuroFS sparsifies the input layer of a sparse neural network (sparse connectivity) until a limited number of input neurons have remained in the network. LassoNet sparsifies the input layer by gradually removing the input neurons until no neurons remain in the input layer. During this removal process, it derives the importance ranking of the features. To have a fair comparison, we get the accuracy of the network having a similar number of input neurons to NeuroFS during this process. Finally, STG exploits a probabilistic gating mechanism in the input layer to select features; if a gate is converged to 0, it is removed from the network; if it is converged to 1, it is selected. As the authors of [3] suggested, when the gate probabilities are not converged to 0/1, a cutoff can be set on the gate/feature probabilities to select features. We use the gate probabilities as the importance of the corresponding features and select the desired number of features (K). We observed in our experiments that when the network is trained for 100 epochs (STG parameter: lambda=0.5), while some gate probabilities might converge to small or large values, they do not converge to exact 0/1. As a result, all input neurons contribute to the network's output. Therefore, we should consider in the results of these experiments that the network trained within the STG algorithm uses all the input features to compute the output.

---

> > ### Author Response · Authors · 2022-11-28
> > **Response to Reviewer Cn4v (2/4)**
> >
> > **1.2. (continued) [Network Accuracy]** Below, you can find the accuracy of each network when selecting K=50 features. Each entry consists of the accuracy of the network (Network-ACC (%)), feature selection accuracy (FS-ACC (%), K=50), and the number of parameters of the corresponding network (#parameters).
> >
> >
> > | Method | Metric | COIL-20 | MNIST | Fashion-MNIST | USPS | Isolet | HAR |
> > |:---:|:---:|:---:|:---:|:---:|:---:|:---:|:---:|
> > | NeuroFS(~20% input features) | FS-ACC | 98.78±0.29 | **95.30±0.41** | **83.78±0.64** | **96.78±0.17** | **92.62±0.40** | 91.46±0.72 |
> > |  | Network-ACC | 31.94±7.88 | 97.22±0.11 | **86.92±0.39** | 95.68±0.22 | **92.31±0.72** | **95.38±0.21** |
> > |  | #parameters (×10^5) | **1.91** | **1.84** | **1.84** | 1.68 | **1.95** | 1.73 |
> > |  LassoNet (~20% input features) | FS-ACC | 97.16±1.06 | 94.46+=10.21 | 82.58±0.10 | 95.94±0.15 | 84.90±0.22 | **93.74±0.39** |
> > |  | Network-ACC | 81.60±8.33 | 84.17±0.11 | 72.29±1.31 | 87.74±0.61 | 84.42±1.31 | 91.34±0.33 |
> > |  | #parameters (×10^5) | 2.64 | 2.16 | 2.16 | **1.11** | 1.99 | **1.68** |
> > |  STG (all input features) | FS-ACC | **99.32±0.40** | 93.20±0.62 | 82.36±0.52 | 96.62±0.34 | 85.82±2.83 | 91.22±1.23 |
> > |  | Network-ACC | **97.92±0.79** | **98.04±0.05** | 86.83±0.34 | **96.19±0.05** | 90.64±0.50 | 90.46±0.79 |
> > |  | #parameters (×10^5) | 30.34 | 27.94 | 27.94 | 22.66 | 26.43 | 25.67 |
> >
> >
> > | Method | Metric | BASEHOCK | Arcene | Prostate_GE | SMK-CAN-187 | GLA-BRA-180 |
> > |:---:|:---:|---|---|---|---|---|
> > | NeuroFS(~20% input features) | FS-ACC | **89.06±2.46** | **76.50±2.55** | **90.50±0.0** | **81.58±1.68** | **80.54±4.96** |
> > |  | Network-ACC | **95.41±0.47** | **86.67±2.36** | 49.27±4.49 | **80.702±1.24** | 72.22±0.00 |
> > |  | #parameters (×10^5) | **2.98** | **4.52** | **3.31** | **7.52** | **16.29** |
> > |  LassoNet (~20% input features) | FS-ACC | 87.18±0.58 | 71.00±2.00 | 88.58±2.35 | 80.52±2.69 | 74.46±4.78 |
> > |  | Network-ACC | 94.90±0.43 | 66.67±1.18 | **87.30±4.49** | 69.30±3.28 | **73.15±1.31** |
> > |  | #parameters (×10^5) | 10.24 | 20.52 | 12.45 | 40.5 | 98.84 |
> > |  STG (all input features) | FS-ACC | 85.12±1.86 | 71.00±2.55 | 84.78±3.55 | 80.25±2.95 | 70.00 ± 4.08 |
> > |  | Network-ACC | 89.57±1.47 | 80.00±6.32 | 84.76±3.56 | 64.21±6.14 | 71.11±4.16 |
> > |  | #parameters (×10^5) | 68.64 | 120.02 | 79.68 | 219.95 | 511.55 |
> >
> > NeuroFS is the best performer in 6 cases in terms of network accuracy compared to LassoNet (with a similar number of input features) and STG (with all input features) while having the smallest number of parameters in most cases (9 out of 11). However, in terms of feature selection performance, it is the best performer in 9 out of 11 cases. This shows that better network accuracy does not necessarily lead to better feature selection performance.
> >
> >
> >
> > **1.3. [Evaluation with other classifiers]** In addition, to show that the result of feature selection is not biased on the chosen classifier, we also measured the classification accuracy using two other widely-used classifiers, i.e., [KNN](https://scikit-learn.org/stable/modules/generated/sklearn.neighbors.KNeighborsClassifier.html) and [ExtraTrees](https://scikit-learn.org/stable/modules/generated/sklearn.ensemble.ExtraTreesClassifier.html) when K=50. Please find the classification accuracy results in the below link.
> >
> >  [**[Click here]: Results with different classifiers**](https://anonymous.4open.science/r/NeuroFS-980D/imgs/ACCs_classifiers.JPG)
> >
> > As can be seen in this table, the overall performance of all methods is consistent across different classifiers in most cases.

---

> > > ### Author Response · Authors · 2022-11-28
> > > **Response to Reviewer Cn4v (3/4)**
> > >
> > > **2. [...example that demonstrates that the model can accurately recover informative features in a noisy environment]**
> > >  We measured the accuracy of the algorithms on the Gisette dataset when selecting K=50 features. As shown below, NeuroFS performs very closely to LassoNet and STG on this dataset while having much fewer FLOPs. Also, NeuroFS outperforms QuickSelection in terms of accuracy while using the same number of FLOPs. It is worth noting that due to multiple rounds of training of LassoNet (more details in [FLOPs count] Section), it has higher training FLOPs than STG, which uses a 3-layer dense network.
> > >
> > > |  | NeuroFS | LassoNet | STG | QS | ICAP | CIFE | RFS | Fisher_Score |
> > > |:---:|---|---|---|---|---|---|---|---|
> > > | Accuracy (%) | 96.5±0.25 | 97.12±0.17 (+0.62) | 96.81±0.34 (+0.31) | 93.86±0.47 (-2.64) | 94.3 (-2.2) | 90.7 (-5.80) | 96.0 (-0.5) | 92.3 (-4.2) |
> > > | FLOPs(×10^12) | 1.09 (×1) | 101.75±17.84 (×93) | 18.01 (×16) | 1.09 (×1) |  |  |  |  |
> > >
> > >
> > >
> > >  **3. [Architecture]**  As discussed in Section 4.1 in the paper, we use a 3-layer MLP with 1000 neurons in each hidden layer for NeuroFS and STG. However, for LassoNet, we used 1-layer MLP with 1000 hidden neurons as also suggested by the authors. We have also tried using a 3-layer MLP for LassoNet. However, it significantly increased the running time, and particularly on large datasets, it exceeded the 12 hours time limit. In addition, in the other cases, it did not lead to significantly different results than LassoNet with 1-layer MLP.
> > >
> > >  **4. [Hyperparameters]**  For NeuroFS, we used fixed hyperparameters for all experiments. Based on our experiments in Section 5.3 in the paper, these values might not be the optimal values for each dataset. However, they mostly lead to good performance on all datasets. For each baseline method, we also used the set of parameters that works well (in terms of test classification accuracy) for various datasets and used in the corresponding implementation or suggested by the authors. Please refer to the response (2/2) to *reviewer 42sy* for more details about the hyperparameters setting. Below, we summarize the performed analysis and discussion about hyperparameters.
> > >
> > > ICAP, CIFE, and Fisher_score have no hyperparameters (except K in ICAP and CIFE, for which we have tried values in [25, 50, 75, 100, 150, 200] for all methods). RFS has a hyperparameter gamma, and we set it to 10; we have discussed the hyperparameter sensitivity of RFS [here](https://anonymous.4open.science/r/NeuroFS-980D/imgs/RFS.JPG).
> > >
> > > For NN-based methods (NeuroFS, LassoNet, STG, and QS), to have a fair comparison, we used similar learning rates (0.01), optimizers (SGD), batch sizes (100, except 20 for datasets with few samples (m<=200)), and training epochs (100). For the parameters specific to each NN-based method: STG hyperparameter: (lambda=0.5, we have analyzed hyperparameter sensitivity of STG [here](https://anonymous.4open.science/r/NeuroFS-980D/imgs/STG.JPG). STG is not very sensitive to this hyperparameter), LassoNet hyperparameter: (M=10 that was suggested by the authors in [2]), QS: (epsilon and zeta. Authors of [5] performed a hyperparameter search and mentioned that this method is not very sensitive to these hyperparameters. Based on their experiment, zeta=0.2 or zeta=0.3 works decently on all datasets. We selected zeta=0.3, which is similar to the zeta value we used in our experiments. For epsilon, which determines the sparsity level/parameter count, we also selected a similar value to our experiments to have a fair comparison in terms of parameter budget).
> > >
> > >
> > >
> > >  **5. [Number of Selected features]** The accuracies in Table 2 in the paper are the average classification accuracy for different K values. We have included the accuracies for each K value separately in Table 5 in the Appendix of the paper. By comparing the results in Tables 2 and 5 in the paper, it can be observed that the average accuracy is representative of the overall performance of each method among the other methods without being biased on a specific K value.
> > >
> > >  **6. [SVM]** We used an SVM classifier with RBF kernel from the [scikit-learn](https://scikit-learn.org/stable/modules/generated/sklearn.svm.SVC.html) library. The other hyperparameters have been set to the default values used in this library. We have clarified this in the updated manuscript.

---

> > > > ### Author Response · Authors · 2022-11-28
> > > > **Response to Reviewer Cn4v (4/4)**
> > > >
> > > > **7. [FLOPs count]** We thank the reviewer for their comment. Below, we discuss the FLOPs count for each method separately.
> > > >
> > > >
> > > > **7.1. [NeuroFS]** First, we highlight that in addition to sparsifying the input layer (gradually removing unimportant input neurons), NeuroFS exploits sparse weight matrices in all layers from the beginning of training. It has a fixed parameter count during training which is around only 3-8% (depending on the dataset as summarized in Table 4 in the paper) of the parameters of the equivalent dense model.
> > > >
> > > > **7.2. [LassoNet]** As the reviewer mentioned, LassoNet sparsifies the input layer by gradually removing the input neurons. However, as we mentioned in Section 5.2 in the paper, this process might need several rounds of training this network depending on the dataset and convergence of the network (Algorithm 1 in [2]). In the paper, we considered one round of training a dense network as a lower bound. For a more detailed comparison, we compute the exact number of FLOPs for feature selection from each dataset. The results are shown below:
> > > >
> > > > |  | Dataset | COIL-20 | MNIST | Fashion-MNIST | USPS | Isolet | HAR |
> > > > |---|---|---|---|---|---|---|---|
> > > > | FlOPs (×10^12) | LassoNet | 4.53±0.00 | 371.02±0.06 | 439.78±0.66 | 10.93±0.03 | 23.61±0.34 | 20.34±0.02 |
> > > > |  | NeuroFS | **0.13** | **6.66** | **6.66** | **0.76** | **0.73** | **0.77** |
> > > >
> > > > |  | Dataset | BASEHOCK | Arcene | Prostate_GE | SMK-CAN-187 | GLA-BRA-180 |
> > > > |---|---|---|---|---|---|---|
> > > > | FlOPs (×10^12) | LassoNet | 21.43±0.01 | 5.84±1.14 | 1.89±0.35 | 4.38±0.11 | 12.60±0.07 |
> > > > |  | NeuroFS | **0.36** | **0.05** | **0.02** | **0.07** | **0.18** |
> > > >
> > > > Due to different convergence, the number of training rounds for training LassoNet might differ for each run (random seed). As a result, the number of FLOPs can be different for each run. Therefore, the FLOPs for LassoNet are presented as mean and standard deviation.
> > > >
> > > > **7.3. [STG]** STG exploits a probabilistic gating mechanism in the input layer to select features; if a gate is converged to 0, it is removed from the network, and if it is converged to 1, it is selected. Therefore, the input layer might be sparse depending on the gate values of the corresponding input neurons, and the other layers are dense. We observed in our experiments when the network is trained for 100 epochs (STG parameters: lambda=0.5), while some gate probabilities might converge to small or large values, they don’t converge to precisely 0/1. Therefore, they contribute to the network, and the input layer of the network was not sparse in these cases. As a result, the FLOPs count is equal to the dense network in these experiments, as shown in Table 4 in the paper.
> > > >
> > > >
> > > >
> > > >
> > > > **[Suggested changes]** We thank reviewer Cn4v for their careful reading and for providing suggestions for improvement. We have incorporated the suggested changes accordingly in the new manuscript.
> > > >
> > > >
> > > >
> > > > ***References:***
> > > >
> > > > [1] Li, Jundong, et al. "Feature selection: A data perspective." ACM computing surveys (CSUR) 50.6 (2017): 1-45.
> > > >
> > > > [2] Lemhadri, Ismael, Feng Ruan, and Rob Tibshirani. "Lassonet: Neural networks with feature sparsity." International Conference on Artificial Intelligence and Statistics. PMLR, 2021.
> > > >
> > > > [3] Yamada, Yutaro, et al. "Feature selection using stochastic gates." International Conference on Machine Learning. PMLR, 2020.
> > > >
> > > > [4] Balın, Muhammed Fatih, Abubakar Abid, and James Zou. "Concrete autoencoders: Differentiable feature selection and reconstruction." International conference on machine learning. PMLR, 2019.
> > > >
> > > > [5] Atashgahi, Zahra, et al. "Quick and robust feature selection: the strength of energy-efficient sparse training for autoencoders." Machine Learning 111.1 (2022): 377-414.

---

> > > > > ### Comment · Reviewer_Cn4v · 2022-12-13
> > > > > **Response to authors**
> > > > >
> > > > > I truly appreciate the author's effort to improve the paper and address all concerns raised by the reviewers. The new experiments are important, and I agree with the authors that the method seems promising in terms of its feature selection capabilities and reduction of FLOPS. However, I still feel that some of my comments were not fully addressed. For example: since the method is a feature selection method, I would expect a systematic evaluation of its capabilities in identifying the informative features in high-dimensional settings. I appreciate the accuracy of GISSETE, but it is important to also report precision/recall/F1/median rank in terms of the ability of the method to recover informative features (this could be done in a synthetic setting; see examples in [1],[2].
> > > > > Regarding the FLOPS evaluation, I appreciate the new results, but I feel that the evaluation of STG + LASSO net is still not performed accurately. Specifically, these methods require tuning the regularization parameter such that FS actually occurs. This can happen when increasing $\lambda$. The FLOPS should be computed by averaging the FLOPS over the training curve, as was performed in [3], which demonstrates substantial FLOPS reduction with a similar technique to STG.
> > > > > *Hyperlinks to Figures of sensitivity do not work.
> > > > >
> > > > > To conclude, overall, I like the paper, and I think the contribution is important. I still think that some aspects could be improved (as mentioned above) before acceptance, but I think these are doable.
> > > > >
> > > > >
> > > > > [1]Chen et al. "Kernel feature selection via conditional covariance minimization."
> > > > > [2] Yamada, Yutaro, et al. "Feature selection using stochastic gates."
> > > > > [3] Louizos et al. "Learning sparse neural networks through $ L\_0 $ regularization."

---

> > > > > > ### Author Response · Authors · 2022-12-19
> > > > > > **Response (2) to Reviewer Cn4v  (1/2)**
> > > > > >
> > > > > > We thank the reviewer for their time reading the response and the revised paper. We are happy to hear that the new experiments are important. Please find below the response regarding the synthetic dataset and FLOPs computation. We hope that we have addressed the reviewer’s concerns sufficiently. If not, we would be happy to further discuss and address any unclear aspects.
> > > > > >
> > > > > > **1. [Synthetic dataset evaluation]**  We thank the reviewer for their suggestion regarding the synthetic dataset evaluation. To perform this experiment, we have used the *make_classification* function from the *sklearn* library. This function generates a synthetic dataset with the desired number of input features, including informative and uninformative features. To simulate a complex scenario, in this experiment, we have created a dataset with 1000 features, including 400 informative features and 600 noisy features, such that the informative features are the first 400 features index.  We have varied the number of samples between {250, 500, 750, 1000, 1500, and 2000} samples. The task is binary classification. Other parameters are set as the default values used in the sklearn library.
> > > > > > We have plotted the classification accuracy for feature selection (K=400), the total number of training FLOPs, and the number of informative features each method founds in the selected subset of features. We have compared the results of NeuroFS with LassoNet and STG (the closely performing methods on the Gisette dataset). The results are shown in the following link.
> > > > > >
> > > > > >  [**Click here: Results on a noisy dataset**](https://anonymous.4open.science/r/NeuroFS-980D/imgs/noisy_dataset.JPG)
> > > > > >
> > > > > > NeuroFS performs better or closely compared to the other methods in terms of accuracy in the low data regime (or high-dimensional regime where the number of samples< number of dimensions) while having much fewer training FLOPs compared to LassoNet and STG. LassoNet finds a higher number of informative features within the selected features and performs well when the number of samples is high. However, it has the highest number of FLOPs, even higher than STG, which uses a 3-layer dense network.

---

> > > > > > > ### Author Response · Authors · 2022-12-19
> > > > > > > **Response (2) to Reviewer Cn4v  (2/2)**
> > > > > > >
> > > > > > > **2. [FLOPs computation]** The reported FLOPs are the total FLOPs during training for each method. ***LassoNet*** updates \lambda in its algorithm to induce feature sparsity in the network. The features are gradually removed from the network during the training path. For computing the FLOPs for LassoNet in the revision, we tracked the active input neurons (neurons with at least one non-zero connection) at each training epoch and took into account the reduction in the number of input neurons (the training path starts with the entire input neurons set and ends up with no active neurons).
> > > > > > >
> > > > > > > ***STG*** generally imposes a gating mechanism on the input layer. The main difference between (Louizos et al 2018) and STG is that in STG, the regularization term in the loss function of STG is only applied to the input features to be suitable for feature selection. On the other hand, (Louizos et al 2018) aim to prune the whole network, and regularization is applied to all of the network’s parameters. Therefore, (Louizos et al 2018) is more efficient in terms of parameter reduction compared to STG, as in STG, we observe sparsity only in the input layer while the rest of the network exploits dense weight matrices. However, in our experiments, the sparsity did not happen during 100 training epochs (we have tried lambda between 0.001, 0.01, 0.5, 1, 10, https://anonymous.4open.science/r/NeuroFS-980D/imgs/STG.JPG ). We used thresholding to select features as suggested by the authors of STG. It should be noted that even finetuning \lambda for each dataset and the desired value of K can be challenging and computationally expensive, while NeuroFS is sparse from scratch without tuning any hyperparameter.
> > > > > > >
> > > > > > > However, to find the lower bound of the number of FLOPs for STG in case sparsity happens in the input layer, we consider the case where only 50 features are active in the input layer of the network for the whole training time (in the actual training, STG starts with the whole feature set and the sparsity might occur in time). The computed FLOPs are summarized below. As can be seen in this table, the number of FLOPs of NeuroFS is lower than the lower bound of STG FLOPs in all considered cases.
> > > > > > >
> > > > > > > |  | Method | COIL-20 | MNIST | Fashion-MNIST | USPS | Isolet | HAR |
> > > > > > > |:---:|:---:|:---:|:---:|:---:|:---:|:---:|:---:|
> > > > > > > | FLOPs (×10^12) | STG | 2.10 | 100.64 | 100.64 | 10.12 | 9.90 | 11.33 |
> > > > > > > |  | STG (50 input, lower bound) | 1.42 | 74.21 | 74.21 | 9.20 | 7.77 | 9.08 |
> > > > > > > |  | NeuroFS | **0.13** | **6.66** | **6.66** | **0.76** | **0.73** | **0.77** |
> > > > > > >
> > > > > > >
> > > > > > > |  | Method | BASEHOCK | Arcene | Prostate_GE | SMK-CAN-187 | GLA-BRA-180 |
> > > > > > > |:---:|:---:|:---:|:---:|:---:|:---:|:---:|
> > > > > > > | FLOPs (×10^12) | STG | 8.21 | 1.44 | 0.49 | 1.97 | 5.52 |
> > > > > > > |  | STG (50 input, lower bound) | 2.46 | 0.25 | 0.13 | 0.18 | 0.22 |
> > > > > > > |  | NeuroFS | **0.36** | **0.05** | **0.02** | **0.07** | **0.18** |
> > > > > > >
> > > > > > >
> > > > > > > ***[Hyperparameter figures]*** Please find the figures in this anonymous Github repository (https://anonymous.4open.science/r/NeuroFS-980D/ ) as *imgs/RFS.JPG* and *imgs/STG.JPG*.

---

### Review · Reviewer_6znL · 2022-11-12

**Summary Of Contributions:**

The paper proposes a neural based feature selection method. The selection is performed during training. At each neural network training epoch, neurons and connections are pruned and activated based on magnitude of weight and gradient values. A scheduling strategy for determining how many neurons and connections are removed/added are proposed, which consists of removal phase (in which # of active neurons decreases) and update phase (in which # of active neurons is fixed). The authors empirically shows the proposed method can maintain high prediction performance even when only a small number of features are used. The performance is compared with several neural based feature selection methods and classical filter methods.

**Audience:**

Yes

**Broader Impact Concerns:**

I have no concern.

**Claims And Evidence:**

Yes

**Requested Changes:**

Critical

- The authors define that the proposed method is an 'embed' approach of the feature selection. On the other hand, experiments only shows performance of SVM with the selected features. Although the authors mention that a non-neural model is intentionally selected for a fair evaluation, this evaluation metric confuses me. Obviously, each embed method optimizes a set of features so that a criterion derived from a specific prediction model. For me, it is quite unclear how it can be justified that only using SVM for evaluating features selected based on neural models. It should be clarified why the SVM based evaluation is valid for the embed feature selection evaluation in more detail, or change the evaluation strategy (such as using original models of each method, or adding prediction models other than SVM to avoid that the result and discussion are unconsiously biased to SVM).

- In introduction, the following two statement are somewhat exaggerated.
  + 'we propose for the first time to use the sparse neural networks to perform supervised feature selection'
  + As a main contribution, 'introduce dynamic neuron pruning and regrowing in the input layer of sparse neural networks during training'.

  Hefler et al., JMLR2021 the authors cited mentioned 'most neuron sparcification schemes can be used for feature selection' (sec2.5), by which I think claiming novelty on applying the neuron sparcification to the supervised feature selection scenario itself is not convincing.

- Is using LassoNet with 1-layer MLP while the proposed method employs a 3-layer MLP fair comparison?

- What criterion is used to determine hyper-parameters of networks?

- The settings of SVM in evaluation metric should have been clarified (kernel, hyper-parameters, and so on).

- In the 'Implementation' paragraph, what does 'similar hyperpameters' mean? Describe precisely.

Not Critical:

- The average accuracy plot of Figure 2 is too small.

**Strengths And Weaknesses:**

Strengths

- The topic is obviously important for a wide range of the community.

- The intuition behind the proposed method is easy to understand and simple to implement.

- The paper is well-organized and easy to follow.

Weaknesses

- Both criteria for the magnitude based pruning of weights and the gradient based activation are quite common. The scheduling strategy that consists of the removal and growth phases can be seen as an original contribution, but the basic strategy is somewhat straightforward. Since the scheduling procedure are heuristically designed, it is not fully motivated why the proposed specific way of the scheduling procedure stably provides good performance.

- Filter based compared methods are somewhat old such as RFS (2010), Fisher score (2011), ICAP (2005) and CIFE (2006). The authors said their main focus is in embed approaches, and so, I guess the authors selected classical methods for filter approaches. However, more modern approaches should have been included for clarifying advantage of the proposed framework as far as filter methods are also used as baselines. For example, well known HSIC based non-linear feature selection strategies can be important baselines, though it is not even mentioned.

---

> ### Author Response · Authors · 2022-11-28
> **Response to Reviewer 6znL (1/2)**
>
> We thank Reviewer 6znL for their careful reading of the manuscript and for providing constructive comments. We appreciate that the reviewer found the **topic important, the method easy to understand and implement, and the paper well-organized and easy to follow**. We try to clarify the comments raised by the reviewer below. We have incorporated the comments raised by the reviewer in the updated version of the paper. We believe that these insightful suggestions have resulted in an improved manuscript. Please let us know if any responses are unclear or if you wish additional changes.
>
>
> **1. [...it is not fully motivated why the proposed specific way of the scheduling procedure stably provides good performance.]**
> As the reviewer mentioned, magnitude-based removal and gradient-based addition of the weights are commonly used policies for updating the structure of sparse neural networks (SNNs) trained within the Dynamic Sparse Training (DST) class of methods. It is discussed and analyzed in-depth in [1] why such an approach performs well. By comparing the gradient growth versus random growth policy in DST in [1], the authors discuss that DST methods improve the gradient flow by updating the sparse connectivity. Particularly, picking the connections with the highest gradient magnitude increases the gradient flow, eventually leading to a decent performance.
>
> Inspired by the weights update policy in DST, we propose to update neurons to remove the uninformative features from the network. Adding the neurons connected to the connections with the highest gradient magnitude improves the gradient flow in the network and keeps the most informative features in the network. As we have shown in Appendix A in the paper, using random weight and neuron addition does not guarantee a stable performance across all datasets. We highlighted the motivation behind our proposed method in the paper. However, we believe that more in-depth studies can be performed, which is out of the scope of the current paper.
>
> **2. [... well-known HSIC-based non-linear feature selection strategies can be important baselines]**
> We considered ICAP, CIFE, and Fisher score as filter methods and RFS as a non-neural network embedded [2] approach (we acknowledge that in some works, RFS is also considered as a filter approach [3]). As the reviewer suggested, we consider two HSIC-based feature selection methods: HSICLasso [4] and HSICLassoVI [5]. The results are summarized below (the non-existing entries for HSICLassoVI are due to the error in the implementation when computing the inverse matrix. We have used the implementation provided by the authors).
>
> [**[Click here] Comparison with HSIC-based feature selection methods**](https://anonymous.4open.science/r/NeuroFS-980D/imgs/HSICLasso.JPG)
>
> As can be seen in this table, NeuroFS outperforms these methods in seven cases while performing very close to the best performer in the other cases (less than a 1% difference in accuracy). We have added the results of this experiment in the new version of the paper in Appendix C.
>
> **3. [...It should be clarified why the SVM-based evaluation is valid for the embed feature selection evaluation in more detail, or change the evaluation strategy]** We used the general approach to evaluate supervised feature selection methods, as suggested in [6] and widely used in the feature selection literature, e.g., [7, 8]. In this approach, we use the classification accuracy of an SVM classifier trained on the subset of the features selected by each method for comparison. The higher the classification accuracy, the better the selected features are [6]. In this way, we can have a fair comparison among various (filter, wrapper, embedded) feature selection algorithms as the evaluation metric is not biased on the training method used by each algorithm. We also measured the classification accuracy using two widely-used classifiers, including KNN and ExtraTrees, besides SVM, used in the submitted version. Please find the classification accuracy results when K=50 below.
>
> [**[Click here] Results with different classifiers**](https://anonymous.4open.science/r/NeuroFS-980D/imgs/ACCs_classifiers.JPG)
>
> As can be seen in this table, the overall performance of all methods is consistent across different classifiers in most cases. We have added the results of this experiment in the new version of the paper in Appendix D.

---

> > ### Author Response · Authors · 2022-11-28
> > **Response to Reviewer 6znL (2/2)**
> >
> > **4. [...applying the neuron sparsification to the supervised feature selection scenario…]** While the works introduced in [9] prune the existing neurons based on their importance, NeuroFS allows the addition of the pruned neurons in case of receiving high gradient magnitude. In other words, in contrast with the greedy removal of neuron pruning algorithms, by introducing the re-addition of neurons to the network, NeuroFS can reverse the wrong greedy choice made in the earlier stages of training. Therefore, we believe that NeuroFS is novel in terms of updating (removing and adding) the input neurons and their dynamic training nature.
> >
> > **5. [LassoNet Architecture]** For LassoNet, we used 1-layer MLP with 1000 hidden neurons, as also suggested by the authors. We have also tried using a 3-layer MLP for LassoNet. However, it significantly increased the running time; particularly on large datasets, it exceeded the 12 hours time limit. In addition, in the other cases, it did not yield significantly different results than LassoNet with 1-layer MLP.
> >
> > **6. [Hyperparameters of networks]** To get a general overview of the performance of the methods, for each method, we used the set of parameters that works well (in terms of the quality of the selected features) for a variety of datasets and used in the corresponding implementation or suggested by the authors. For NN-based methods (NeuroFS, LassoNet, STG, and QS), to have a fair comparison, we used similar learning rates (0.01), optimizers (SGD), batch sizes (100, except 20 for datasets with few samples (m<=200)), and training epochs (100). For the parameters specific to each NN-based method: STG hyperparameter: (lambda=0.5), LassoNet hyperparameter: (M=10), QS: (epsilon = 30, zeta = 0.3). Please also refer to the response (2/2) to *reviewer 42sy* for more details about the hyperparameters setting.
> >
> >
> > **7. [SVM]** We used an [SVM classifier](https://scikit-learn.org/stable/modules/generated/sklearn.svm.SVC.html) with RBF kernel from the scikit-learn library. The other hyperparameters have been set to the default values used in this library. We clarified this in the updated version of the paper.
> >
> > **8. [...similar hyperparameters]** We have updated the hyperparameter details in the implementation paragraph in the new manuscript.
> >
> >
> >
> >
> > **References:**
> >
> > [1] Evci, Utku, et al. "Gradient flow in sparse neural networks and how lottery tickets win." Proceedings of the AAAI Conference on Artificial Intelligence. Vol. 36. No. 6. 2022.
> >
> > [2] Zhang, Rui, et al. "Feature selection with multi-view data: A survey." Information Fusion 50 (2019): 158-167.
> >
> > [3] Solorio-Fernández, Saúl, J. Ariel Carrasco-Ochoa, and José Fco Martínez-Trinidad. "A review of unsupervised feature selection methods." Artificial Intelligence Review 53.2 (2020): 907-948.
> >
> > [4] Yamada, Makoto, et al. "High-dimensional feature selection by feature-wise kernelized lasso." Neural computation 26.1 (2014): 185-207.
> >
> > [5] Koyama, Kazuki, et al. "Effective Nonlinear Feature Selection Method based on HSIC Lasso and with Variational Inference." International Conference on Artificial Intelligence and Statistics. PMLR, 2022.
> >
> > [6] Li, Jundong, et al. "Feature selection: A data perspective." ACM computing surveys (CSUR) 50.6 (2017): 1-45.
> >
> > [7] Lemhadri, Ismael, Feng Ruan, and Rob Tibshirani. "Lassonet: Neural networks with feature sparsity." International Conference on Artificial Intelligence and Statistics. PMLR, 2021.
> >
> > [8] Balın, Muhammed Fatih, Abubakar Abid, and James Zou. "Concrete autoencoders: Differentiable feature selection and reconstruction." International conference on machine learning. PMLR, 2019.
> >
> > [9] Hoefler, Torsten, et al. "Sparsity in Deep Learning: Pruning and growth for efficient inference and training in neural networks." J. Mach. Learn. Res. 22.241 (2021): 1-124.

---

### Review · Reviewer_42sy · 2022-11-24

**Summary Of Contributions:**

This paper proposes a feature selection method for sparse three layer perceptrons.
The proposed method consists of successive pruning of the input features as well as regrowth.
For pruning, the authors proposed using the magnitude of the connection weights related to each of active features.
For regrowth, the authors proposed using the magnitude of the gradient related to each of inactive features.
The proposed method then combines these pruning are regrowth steps with carefully tuned scheduling so that the number of total active features to decrease during the training.
In the experiments, the authors showed that the proposed method outperformed some of the existing feature selection methods for neural networks.

**Audience:**

Yes

**Broader Impact Concerns:**

There is no ethical concern.

**Claims And Evidence:**

No

**Requested Changes:**

#### Request 1.
Please describe the difference and the advantage of the proposed method from the prior methods that also use pruning and regrowth.
Why and when the proposed pruning/regrowth strategies are considered better than the previous ones?

#### Request 2.
Please describe the choice of hyperparameters used in the baseline methods in the experiments.
I wonder whether the authors used the implementations with their default hyperparameters, the ones far from optimal.

**Strengths And Weaknesses:**

### Strong aspects

I could not find any strong aspects of the paper.
Currently, to me, the proposed method looks like a special case of RigL Evci et al. (2020) with pruning and regrowth applied only to the first layer (see Weakness 1).


### Weak aspects

#### Weakness 1: There is a critical concern regarding the novelty of the paper.
The idea of the proposed method is on the dynamic pruning and regrowth of the features during training.
As the authors mention in the paper, such an idea itself is not novel.
For example, Mocanu et al. (2018) considers the similar pruning criterion with random regrowth.
RigL Evci et al. (2020) considers the similar strategy as the proposed method; the pruning based on the weight magnitudes and the regrowth based on the gradient magnitudes (not only in the input layer, but also the entire networks).
With these prior studies in mind, the question is what is the advantage of the proposed strategy over these approaches.
In the current paper, the authors merely states that "most of these methods suffer from over-parameterization, which leads
to high computational costs, particularly on high-dimensional datasets" without clear evidence (see Weakness 2 below).
There is no discussion on why the proposed method is more advantageous over the prior methods, e.g., why and when the proposed pruning/regrowth strategies can be considered better than the previous ones.
Currently, to me, the proposed method looks like a special case of RigL Evci et al. (2020) with pruning and regrowth applied only to the first layer.

#### Weakness 2: There is a critical concern regarding the novelty of the paper.
The authors claimed that "most of these methods suffer from over-parameterization, which leads to high computational costs, particularly on high-dimensional datasets" based on their experimental results.
The concern here is on the tuning of these prior methods.
According to Section 4.1, the authors said "we used the implementation provided by the authors" without specifying how their hyperparameters are selected.
Here, I wonder whether the authors used the implementations with their default hyperparameters, the ones far from optimal.
If this is the case, all the claims in the paper based on the experiments can be meaningless.

---

> ### Author Response · Authors · 2022-11-28
> **Response to Reviewer 42sy (1/2)**
>
> We thank Reviewer 42sy for reading and reviewing our manuscript. We clarify the comments raised by the reviewer below. Please let us know if any responses need to be clarified or if you wish additional changes.
>
>
>
>  **1. [Novelty]**
> We respectfully disagree with the reviewer's point of view that the "idea itself is not novel." One of the main differences consists in the problem addressed itself. While Mocanu et al. (2018) and Evci et al. (2020) focus on the typical classification task, our work focuses on the feature selection task. Moreover, Magnitude-based removal and gradient-based addition of the weights are commonly used policies for updating the structure of sparse neural networks (SNNs) trained with the Dynamic Sparse Training (DST) class of methods [3]. We have also cited [5] and mentioned that the regrowth idea is inspired by RigL [5]. Another difference between these works and ours is that while [5] uses the DST concepts for **only weight addition and removal, our work introduces dynamic neurons addition and removal** at each epoch to **perform accurate feature selection**.
>
>
> It is discussed in-depth in [4] that picking the connections with the highest gradient magnitude in RigL results in an increase in the gradient flow, which eventually leads to a decent performance. Inspired by this and the weights update in dynamic sparse training, we propose to update neurons and remove the uninformative features from the network. Adding the neurons connected to the connections with the highest gradient magnitude improves the gradient flow in the network and keeps the most informative features in the network.
>
> In short, **the main novelty of our work is proposing the neuron update mechanism to perform feature selection efficiently.**

---

> > ### Author Response · Authors · 2022-11-28
> > **Response to reviewer 42sy (2/2)**
> >
> >  **2. [Hyperparameters]**
> >  We thank the reviewer for raising this constructive comment. Based on the reviewer's comment, we examined the choice of hyperparameters for the baselines. Below, we present more details about the chosen hyperparameters for each method. We have incorporated these changes in the new manuscript. While the claims in our paper remained unchanged, these changes led to an improved manuscript.
> >
> > For NeuroFS, we used fixed hyperparameters for all experiments. Based on our experiments in Section 5.3 in the paper, these values might not be the optimal values for each dataset. However, they mostly lead to good performance on all datasets. For each baseline method, we also used the set of parameters that works well for various datasets and used in the corresponding implementation or suggested by the authors.
> >
> >  For ICAP, CIFE, RFS, and Fisher_score, we used the implementation and settings provided in [1], which are widely used in the literature [2]. In the following, we examine the settings of each of these methods separately:
> >
> >
> >   - *CIFE, ICAP*: These methods have no hyperparameters to tune except the K (number of features to select). We have tried K in [25, 50, 75, 100, 150, 200] for all methods in our experiments.
> >   - *Fisher_score* has no hyperparameters.
> >   - *RFS* has one hyperparameter gamma. We initially used gamma=0.1 in our experiments which were used in the [scikit-feature](https://github.com/jundongl/scikit-feature/blob/master/skfeature/example/test_RFS.py) library [1]. Below, we show the performance of the algorithm when changing gamma in [0.01, 0.1, 0.5, 1, 10].
> >  [**[Click here] RFS hyperparameter sensitivity**](https://anonymous.4open.science/r/NeuroFS-980D/imgs/RFS.JPG)
> > Based on these results, RFS has the best performance when gamma=10. It should be noted that the experiments on SMK take longer than 12 hours when gamma is in [1, 10]. Therefore, we did not include it in this plot.
> > We have incorporated the new results for RFS(gamma=10) in the new version of the paper. However, the claims and conclusions made in the paper remain unchanged. As can be seen in Figure 3 in the new manuscript, NeuroFS is the best performer in both low and high-dimensional datasets. In addition, NN-based methods work better in general compared to non-NN-based methods.
> >
> >  For NN-based methods (NeuroFS, LassoNet, STG, and QS), to have a fair comparison, we used similar learning rates (0.01), optimizers (SGD), batch sizes (100, except 20 for datasets with few samples (m<=200)), and training epochs (100). Below we explain the choice of hyperparameters for each baseline method separately:
> >
> >   - *LassoNet* has a hyperparameter M. The authors mentioned in their paper [2] that M=10 works well across different datasets. Therefore, we also used M=10 in our experiments.
> >   - *STG* has a hyperparameter lambda [7]. In our experiments, we used lambda=0.5. Below, we show the performance of STG when changing lambda in [0.001, 0.01,  0.5,  1, 10].
> >   [**[Click here] STG hyperparameter sensitivity**](https://anonymous.4open.science/r/NeuroFS-980D/imgs/STG.JPG)
> >   As can be seen in this plot, STG is not very sensitive to this hyperparameter, and the original value chosen in the paper (lambda=0.5) seems to be a good value.
> >
> >   - *QuickSelection*: QS has two hyperparameters: epsilon and zeta. Authors of [6] performed a hyperparameter search and mentioned that this method is not very sensitive to these hyperparameters. Based on their experiment, zeta=0.2 or zeta=0.3 works decently on all datasets. We selected zeta=0.3, which is similar to the zeta value we used in our experiments. For epsilon, which determines the sparsity level/parameter count, we also selected a similar value to our experiments to have a fair comparison in terms of parameter budget.
> >
> >
> >
> >  ***References:***
> >
> > [1] Li, Jundong, et al. "Feature selection: A data perspective." ACM computing surveys (CSUR) 50.6 (2017): 1-45.
> >
> > [2] Lemhadri, Ismael, Feng Ruan, and Rob Tibshirani. "Lassonet: Neural networks with feature sparsity." International Conference on Artificial Intelligence and Statistics. PMLR, 2021.
> >
> > [3] Hoefler, Torsten, et al. "Sparsity in Deep Learning: Pruning and growth for efficient inference and training in neural networks." J. Mach. Learn. Res. 22.241 (2021): 1-124.
> >
> > [4] Evci, Utku, et al. "Gradient flow in sparse neural networks and how lottery tickets win." Proceedings of the AAAI Conference on Artificial Intelligence. Vol. 36. No. 6. 2022.
> >
> > [5] Evci, Utku, et al. "Rigging the lottery: Making all tickets winners." International Conference on Machine Learning. PMLR, 2020.
> >
> > [6] Atashgahi, Zahra, et al. "Quick and robust feature selection: the strength of energy-efficient sparse training for autoencoders." Machine Learning 111.1 (2022): 377-414.
> >
> > [7] Yamada, Yutaro, et al. "Feature selection using stochastic gates." International Conference on Machine Learning. PMLR, 2020.

---

> > > ### Comment · Reviewer_6znL · 2022-12-23
> > > **.**
> > >
> > > Thank you for responding to my comments. In particular, I appreciate the new results on different classifiers.
> > >
> > > > While the works introduced in [9] prune the existing neurons based on their importance, NeuroFS allows the addition of the pruned neurons in case of receiving high gradient magnitude
> > >
> > > Even after reading the reply, I still think the claim 'first time to use the sparse neural networks to perform supervised feature selection' is obviously overemphasized. This should be rephrased to more moderate descriptions.
> > >
> > > > In this way, we can have a fair comparison among various (filter, wrapper,embedded) feature selection algorithms as the evaluation metric is not biased on the training method usedby each algorithm.
> > >
> > > Since the authors newly added different classifiers, my original concerns on this issue are resolved. However, the above statement itself is not convincing. As I mentioned in the first review, an embed feature selector select features based on a criterion of its 'embedded' algorithm (in this case, a neural network). An extreme example is that it is obviously not optimal to select features for a linear model by using a neural network feature selector. In other words, this means that a classifier may prefer a specific feature selector to other selectors (e.g., SVM might prefer an SVM based embed feature selector). My point was that, in this sense, only using a single classifier (SVM) can make the discussion about the results specific to the employed classifier. That's said, the authors added the classifiers in the revision, by which this concern is already mitigated.

---

> > > > ### Author Response · Authors · 2023-01-05
> > > > **Response to reviewer 6znL**
> > > >
> > > > We thank the reviewer for their time reading our response and the revised paper. We will update the claim in the revised version of the paper. We are happy that the other responses, particularly the other classifiers, were sufficient.

---

### Decision · Action_Editors · 2023-01-05

**Recommendation:** Accept with minor revision

**Comment:**

In this paper, the authors propose a neural-based feature selection method called NeuroFS, where feature selection for NN is an important machine learning research topic. More specifically,  NeuroFS gradually prunes the uninformative features from the input layer of a sparse neural network trained from scratch.

The critical concern for this paper is the difference from the RigL (Evci 2020). As pointed out by Reviewer 42sy, "the proposed method looks like a special case of RigL Evci et al. (2020) with pruning and regrowth applied only to the first layer." Then, the authors respond to the question as "While Mocanu et al. (2018) and Evci et al. (2020) focus on the typical classification task, our work focuses on the feature selection task." I agree that Evci et al. (2020) focus on training sparse neural networks. However, RigL can be potentially used for feature selection, as written in its appendix Fig 7 (https://arxiv.org/pdf/1911.11134.pdf). Then, the important point of this paper is that NeuroFS can select a certain number of features, while RigL may not select. I agree that this is an important property for feature selection.

However, it remains a crucial question; how feature selection technique is significant over RigL + simple heuristic-based feature selection algorithm? For example, after we train RigL, we can select features by ranking feature importance (magnitude of features in the first layer) and selecting top-K features.

Thus, I request the authors to the comparison with RigL for the camera-ready version. Adding RigL for all the experiments is preferable. The similar experiments in the section "C Comparison with HSICLasso-based Feature Selection Methods" is acceptable.

**Audience:**

Feature selection for neural networks is an important topic, particularly for high-dimensional data.

**Claims And Evidence:**

The claims made in the submission are mostly supported.

---

> ### Author Response · Authors · 2023-02-02
> **Thank you for the reviews!**
>
> We would like to thank all reviewers and the action editor for your precious time in reviewing our manuscript and providing feedback. We believe insightful and constructive feedback helped us to improve the paper. We have uploaded a camera-ready version with the following revision. We have added the comparison with RigL as Appendix D in the paper.

---

> > ### Comment · Action_Editors · 2023-02-02
> > **Could you add a discussion of RiGL in Appendix D?**
> >
> > Dear authors,
> >
> >  Thank you for uploading the camera-ready version. I saw the results of RigL in Appendix D and it shows that the proposed method performs better than RigL. This is great. However, it is still not sure why the proposed method is better than RigL. Also, the details of the RigL setup are missing.
> >
> > So, could you please add more detail on how to run feature selection with the RigL models? Also, please add the discussion of why the proposed method can outperform RigL.
> >
> > Thanks.

---

> > > ### Author Response · Authors · 2023-02-06
> > > **Manuscript updated**
> > >
> > > We would like to thank the editor for their feedback.  As requested, we have added more details regarding the settings and the feature selection using RigL. We have also updated the experiments and the discussion in Appendix D. Please let us know if further changes/experiments are required.
> > >
> > > Regards,
> > > Authors